

# A combined observational and modelling approach to evaluate aerosol-cirrus interactions at high and mid-latitudes

Elena De La Torre Castro[1], Christof G. Beer[1], Tina Jurkat-Witschas[1], Daniel Sauer[1], Mattia Righi[1], Johannes Hendricks[1], and Christiane Voigt[1, 2]

[1]Deutsches Zentrum für Luft- und Raumfahrt, Institut für Physik der Atmosphäre, Oberpfaffenhofen, Germany
[2]Johannes Gutenberg Universität Mainz, Institut für Physik der Atmosphäre, Mainz, Germany

**Correspondence:** Elena De La Torre Castro (Elena.delaTorreCastro@dlr.de)

**Abstract.** Aerosol-cirrus interactions remain a major source of uncertainty in climate models due to the complex interplay of aerosol properties, ice nucleation pathways, and atmospheric conditions. In this study, we investigate the drivers of observed differences in cirrus microphysical properties between high and mid-latitudes from the CIRRUS-HL campaign by combining observations with simulations from a global aerosol-climate model. While mid-latitude cirrus exhibit median ice crystal number concentrations ($N_{\mathrm{ice}}$) one order of magnitude higher than those at high latitudes, aerosol concentrations ($N_{\mathrm{aer}}$) integrated across several sizes ranges are similar at cirrus altitudes in both regions. By coupling the model output with backward trajectories, we attribute the differences in $N_{\mathrm{ice}}$ to diverse influences of specific ice-nucleating particle (INP) types with distinct freezing efficiencies rather than to total aerosol or INP number concentrations. Mineral dust plays a dominant role in cirrus formation at mid-latitudes, while aviation-emitted black carbon may contribute to high-latitude cirrus assuming it acts as an efficient INP. The model reproduces aerosol observations reasonably well but underestimates $N_{\mathrm{aer},\ D>250\ \mathrm{nm}}$ at high latitudes near 300 hPa. At mid-latitudes, it overestimates $N_{\mathrm{ice}}$ at temperatures above 220 K, primarily due to an overestimation of the concentration of ice crystals detrained from convective clouds. Incorporating a size parametrization for convective ice crystals derived from CIRRUS-HL measurements significantly reduces this bias, which represents a fundamental improvement to the cloud scheme. These findings highlight the value of integrating observations with model simulations to interpret field measurements and improve the representations of cirrus clouds in global models.

## 1 Introduction

Cirrus clouds play a crucial role in the Earth's energy budget and climate system. Their net radiative effect depends on the balance between shortwave and longwave radiation, which in turn is influenced by factors such as cloud location, season, solar zenith angle, surface conditions, and, importantly, cirrus microphysical properties (Zhang et al., 1999; Stephens, 2005; Wendisch et al., 2007). These properties are influenced by complex interactions involving aerosol concentrations and chemical composition, affected by anthropogenic influences, as well as by thermodynamic (temperature and relative humidity) and dynamic conditions (updraft velocity) (Hendricks et al., 2011; Patnaude and Diao, 2020; Maciel et al., 2023). Consequently,



understanding aerosol-cirrus interactions is essential for improving climate models and reducing uncertainties in climate predictions (Forster et al., 2021).

Aerosols, originating from both natural and anthropogenic sources, drive the formation of cirrus clouds through two primary pathways: homogeneous and heterogeneous ice nucleation. Homogeneous nucleation occurs when aqueous solution aerosol droplets freeze spontaneously at temperatures below $-38\,°C$ and high relative humidity over ice ($RH_{ice}$, above $140-150\%$), producing large ice crystal numbers (Kärcher and Lohmann, 2002a, b). On the other hand, heterogeneous nucleation involves the presence of insoluble aerosol particles (or ice-nucleating particles, INPs) that facilitate ice crystal formation at higher tem-

peratures and lower supersaturations, leading to lower ice crystal concentrations (Kärcher and Lohmann, 2003). The interplay between these nucleation processes and the ambient meteorological conditions ultimately shape the microphysical properties of cirrus, influencing their radiative properties and lifetime (Baran, 2004; Patnaude and Diao, 2020; Kärcher et al., 2022).

While heterogeneous nucleation dominates in many regions of the globe and across different altitudes in the upper troposphere, especially over the polluted Northern Hemisphere (Cziczo et al., 2013), homogeneous nucleation is believed to occur

mainly in the uppermost troposphere and lower stratosphere. Homogeneous nucleation is typically related to high updrafts over mountains, associated with jet streams, or during winter at high latitudes where INP availability is limited (Gasparini and Lohmann, 2016). Despite considerable progress in observational, laboratory, and modelling studies, significant uncertainties remain regarding heterogeneous nucleation processes and the concentration and freezing properties of INPs, (Kanji et al., 2017), contributing largely to the uncertainties in the climate impact contribution of aerosol-cirrus interactions.

Understanding and accurately representing aerosol-cloud interactions is particularly critical in the Arctic to constrain their effects in the phenomenon of Arctic Amplification (Mauritsen et al., 2011; Schmale et al., 2021; Moser et al., 2023; Wendisch et al., 2024). In situ observations have reported distinct differences in cirrus microphysical properties between high and mid-latitudes (Wolf et al., 2018; De La Torre Castro et al., 2023). In these studies, a low availability of INPs at high latitudes was hypothesized to be the explanation for those differences.

Mineral dust is widely recognized as the most effective and abundant INP type in the upper troposphere (Möhler et al., 2006; Froyd et al., 2022). However, the role of aviation-emitted black carbon particles remains under discussion (Righi et al., 2021; Kärcher et al., 2023; Groß et al., 2023). Emitted directly at cruise altitudes, where cirrus often form, black carbon from aviation (also called soot) particles are particularly abundant in the mid-latitudes of the Northern Hemisphere and could potentially influence cirrus properties (Urbanek et al., 2018; Groß et al., 2023). However, recent measurements indicate that

aviation black carbon might nucleate only at relative humidities close to or above those required for homogeneous freezing (Testa et al., 2024). Moreover, other aerosol species, such as glassy organics or crystalline ammonium sulfate, could have a potential effect on cirrus formation at cold temperatures. However, most global climate models do not include them due to significant uncertainties in their freezing potential, arising from the limited number of observations and studies (Abbatt et al., 2006; Ladino et al., 2014; Beer et al., 2022).

In this study, we aim to address the open questions raised in De La Torre Castro et al. (2023) regarding aerosol-cirrus interactions during the Cirrus in High-Latitudes (CIRRUS-HL) campaign (Jurkat-Witschas et al., 2025). Specifically, we investigate to what extent differences in aerosol particle concentration and composition between high and mid-latitudes explain the observed



variations in ice crystal number concentrations. To this end, we develop a methodology that integrates simulations from the
EMAC model (ECHAM/MESSy Atmospheric Chemistry; Jöckel et al., 2010) coupled with the aerosol microphysics submodel
MADE3 (Modal Aerosol Dynamics model for Europe adapted for global applications, third generation; Kaiser et al., 2019)
with in situ measurements performed during CIRRUS-HL. Our approach jointly evaluates aerosol and ice properties in both
high- and mid-latitude regions and provides a rigorous comparison between observations and specific simulations during the
campaign period. Furthermore, by combining model data with backward trajectories of sampled air masses, we investigate ice
nucleation processes and assess the potential roles of mineral dust, aviation black carbon, and ground-sourced black carbon
throughout the cirrus lifecycle. Based on this analysis, we identify key drivers of the observed cirrus properties and implement
targeted changes in the model, leading to an improved representation of cirrus clouds. Our methodology improves the inter-
comparison between model and observations from the study by Righi et al. (2020) by applying model data directly extracted
along the flight tracks. The Righi et al. (2020) study evaluated the EMAC model with the aerosol submodel MADE3 using a
comprehensive ice cloud data set from 18 aircraft campaigns (Krämer et al., 2009; Krämer et al., 2016) covering the period
from 1996 to 2005. In addition, the study included an intercomparison of the model with observations from the ML-CIRRUS
campaign (Voigt et al., 2017).

An earlier version of this work was included in the PhD thesis by De La Torre Castro (2024). Section 2 provides a brief
overview of the cloud and aerosol in situ measurements from CIRRUS-HL. Section 3 details the configuration of the EMAC
model with the aerosol submodel MADE3 and the methodology used in this study. In Sect. 4, we present an overview of the
CIRRUS-HL observations (Sect. 4.1), an intercomparison between the model and observational data for ice and aerosol parti-
cles (Sect. 4.2), an analysis of nucleation mechanisms and atmospheric processes influencing cirrus microphysical properties
during the CIRRUS-HL campaign (Sect. 4.3), and an additional analysis of the role of convection in the model estimation of
total ice crystal number concentrations (Sect. 4.4). Finally, Sect. 5 summarizes the key findings and offers final remarks.

## 2 Cirrus and aerosol in situ data: CIRRUS-HL observations

In this study, we focus on the cirrus and aerosol properties observed during the CIRRUS-HL campaign in summer 2021 with the
High Altitude and LOng range research aircraft (HALO) (Jurkat-Witschas et al., 2025). The mission included 24 flights under
various meteorological conditions, covering a wide region from mid- to high latitudes over Central Europe, the North Atlantic,
and the Arctic, reaching up to 76°N. The synoptic situation during the campaign was characterized by unstable conditions with
enhanced moisture (Krüger et al., 2022), leading to frequent thunderstorms and hail over Western and Central Europe. Two
flights were specifically targeted in convective systems (Tomsche et al., 2024) and were excluded from our analysis (except in
Sect. 4.4). In total, approximately 25 h of in situ ice particle data and 104 h of in situ aerosol data were collected.

### 2.1 Cloud particle data

The cloud particle measurements were obtained using the Cloud Combination Probe (CCP) and the Precipitation Imaging
Probe (PIP), mounted on the HALO aircraft. Together, these instruments provide cloud particle size distributions from 2 to



6400 µm. A detailed description and characterization of the instrumentation and data set are available in De La Torre Castro et al. (2023) and De La Torre Castro (2024).

The Cloud Droplet Probe (CDP), part of the CCP, is a forward scattering probe that sizes and counts particles in the 2-50 µm range based on Mie theory (Lance et al., 2010; Baumgardner et al., 2011, 2017; Klingebiel et al., 2015; Weigel et al., 2016). Particle number concentrations were determined at a 1-second time resolution. Based on inter arrival time analysis
(Field et al., 2003, 2006), no evidence of shattering artifacts was identified in the data, attributed to the aerodynamic tip design of the instrument arms and the typically low particle concentration in cirrus (McFarquhar et al., 2007, 2011; Korolev et al., 2013).

The Cloud Imaging Probe (CIPgs) and PIP are optical array probes that record two-dimensional images of particles by capturing their shadow projections on a laser-illuminated 64-diode array, as the particle passes the sample area of the instrument
(Baumgardner et al., 2001; Weigel et al., 2016; Voigt et al., 2017). The CIPgs records grayscale images with four intensity levels, improving the shape analysis, while the PIP operates in monoscale. To ensure consistency, a $50\%$ grayscale threshold was applied to the CIPgs image data (O'Shea et al., 2019, 2021). The CIPgs covers nominal particle size ranges from $15$ to $960$ µm with a pixel resolution of $15$ µm, while the PIP detects particles from $100$ to $6400$ µm with $100$ µm pixel resolution. Particle diameters were determined using the minimum enclosing circle method (Heymsfield et al., 2002), and the data were
evaluated applying the all-in method (Knollenberg, 1970; Heymsfield and Parrish, 1978). One-pixel images and other artifacts were identified and excluded from the analysis.

To obtain a complete particle size distribution, data from the three instruments were combined. The key microphysical cirrus parameters analysed in this study include ice crystal number concentration ($N_{\mathrm{ice}}$) and ice water content ($IWC$), both derived from the combined data set. We calculate the $IWC$ via the mass-dimension relationship $m = a \cdot D^b$ (in cgs units) with
coefficients $a = 0.00528$ and $b = 2.1$, as proposed by Heymsfield et al. (2010). We estimate an overall counting and sizing uncertainty of $20\%$ and $50\%$, respectively (De La Torre Castro, 2024). Only measurements taken at ambient temperatures below $-38\,^\circ$C were considered to ensure fully glaciated clouds.

## 2.2 Aerosol data

The aerosol measurement system utilized by the German Aerospace Center (DLR) aboard the HALO aircraft is known as the
AMETYST (Aerosol MEasuremenT sYSTem). Particles with diameters larger than $12$ nm are measured using a Condensation Particle Counter (CPC; model Grimm SkyCPC 5.410), while an Optical Particle Counter (OPC; model SkyOPC 1.129) is employed for larger particle sizes ($250 < D[\mathrm{nm}] < 3000$) (Minikin et al., 2012). Both instruments were adapted by DLR for aircraft applications (Voigt et al., 2022; Dischl et al., 2024). The operation principle is based on the detection of forward scattered light from a laser beam, similar to the principle of forward scattering cloud probes.
The OPC not only provides information on the total number of sampled particles, but also allows for the derivation of particle sizes, with particle counts sorted into various size channels (Walser et al., 2017). However, particles falling within the CPC size range are not directly detectable. Instead, the instrument creates a highly supersaturated environment using the vapor of a working substance, typically butanol, to activate all particles and grow them to sizes detectable by the instrument's optical block





(Minikin et al., 2012). A sigmoid function representing the counting efficiency dependent on size enables the determination of the cut-off diameter, which represents the size of particles counted with an efficiency of at least $50\%$. It mainly depends on the temperature difference between the saturator and condenser, adjustable through laboratory calibrations accordingly (Walser, 2017). However, when using an aircraft as measurement platform, the cut-off diameter is further influenced by losses in the sampling lines from the inlet to the instrument.

The AMETYST system comprises three CPCs. CPC0 and CPC3 for total aerosol concentrations, with nominal cut-off diameters of $\approx 7\,\mathrm{nm}$ and $\approx 18\,\mathrm{nm}$, respectively. However, the real cut-off diameter of CPC0 increases with decreasing pressure due to inlet losses. Andreae et al. (2018) reported cut-off diameters of $11.2\,\mathrm{nm}$ at $500\,\mathrm{hPa}$ and $18.5\,\mathrm{nm}$ at $150\,\mathrm{hPa}$ for the AMETYST system. The inlet system was improved for the CIRRUS-HL campaign, and a smaller cut-off diameter of $\approx 12\,\mathrm{nm}$ is assumed for these measurements (Wolf, 2023). Additionally, CPC2 is dedicated to measuring non-volatile particles larger than approximately $14\,\mathrm{nm}$. Non-volatile particles can be measured by positioning a thermodenuder upstream of the CPC (Fierz et al., 2007). By adjusting the set temperature to the evaporation temperature of the volatile material, only non-volatile particles are retained. The upper diameter limit of the CPC is governed by the inlet characteristics (specifically, the HASI isokinetic inlet), estimated to fall between $1.5$ and $3\,\mathrm{\mu m}$. However, this upper limit is not particularly relevant for integrated number concentration measurements as the lower particle size mode dominates, offsetting the contribution of larger particles to the total number concentration.

## 3 Model description and combination with in situ observations

### 3.1 Model description

The simulations for this study have been performed with the global climate model EMAC. EMAC is a global numerical chemistry and climate simulation system and includes various submodels that describe tropospheric and middle-atmosphere processes. It uses the second version of MESSy to connect multi-institutional computer codes. The core atmospheric model is the ECHAM5 (fifth-generation European Centre Hamburg) general circulation model (Roeckner et al., 2006). In this study we apply EMAC (ECHAM5 version 5.3.02, MESSy version 2.54) in the T42L41DLR configuration with spherical truncation of T42 (corresponding to a horizontal resolution of about $2.8° \times 2.8°$ in latitude and longitude) and 41 non-equidistant vertical layers from the surface to $10\,\mathrm{hPa}$. The simulated time period analysed in this study covers the duration of the CIRRUS-HL campaign taking place in June and July 2021. The model time step of the simulation has a length of 15 minutes and we use a time interval of 1 hour for the output of model data along backward trajectories. The simulations have been performed in nudged mode, i.e. the model meteorology (temperature, winds and logarithm of the surface pressure) is relaxed towards ECMWF reanalysis data (ERA-Interim; Dee et al., 2011) for the simulated time period.

The aerosol microphysics are simulated using the submodel MADE3 that considers different aerosol species in nine log-normal modes representing different particle sizes and mixing states. Each of the MADE3 Aitken-, accumulation-, and coarse-mode size ranges includes three modes for different particle mixing states: particles fully composed of water-soluble components, particles mainly composed of insoluble material (i.e. insoluble particles with only very thin coatings of soluble material),



**Table 1.** Freezing properties of ice-nucleating particles (INPs) in the cirrus regime assumed in this study, i.e. critical supersaturation $S_c$ and activated fraction $f_{\mathrm{act}}$ at the freezing onset. $S_i$ is the supersaturation with respect to ice. In addition to $f_{\mathrm{act}}$ values representative of the freezing onset, $f_{\mathrm{act}}$ values at about $S_i = 1.4$ are used for the analysis in Fig. 6.

| Freezing mode | | $S_c$ | $f_{\mathrm{act}}$ at onset $S_c$ | $f_{\mathrm{act}}$ at $S_i = 1.4$ | Reference |
|---|---|---|---|---|---|
| DU deposition | $T \leq 220\,\mathrm{K}$ | 1.10 | $\exp[2\,(S_i - S_c)] - 1$ | 0.822 | Möhler et al. (2006) |
| | $T > 220\,\mathrm{K}$ | 1.20 | $\exp[0.5\,(S_i - S_c)] - 1$ | 0.105 | |
| BC aviation | | $1.20^a$ | 0.001 | 0.01 | Righi et al. (2021) |
| DU immersion | | 1.35 | 0.01 | 0.1 | Kulkarni et al. (2014) |
| BC | | 1.40 | 0.0025 | 0.0025 | Righi et al. (2021) |

[a] This is an assumption of efficient ice formation by aviation soot (see the text)

and mixed particles (i.e. soluble material with inclusions of insoluble particles). A detailed description of MADE3 and its application and evaluation as part of EMAC are presented in Kaiser et al. (2014, 2019).

In this study, EMAC is employed in a coupled configuration featuring a two-moment cloud microphysical scheme based on Kuebbeler et al. (2014), which parametrizes aerosol-driven ice formation in cirrus clouds following Kärcher et al. (2006). The Kärcher et al. (2006) scheme considers different ice formation mechanisms that compete for the available supersaturated water vapour, i.e. homogeneous freezing of solution droplets, deposition and immersion nucleation induced by INPs, and the growth of preexisting ice crystals. For simplicity the scheme does not represent the entire freezing spectrum (from the freezing onset to the homogeneous freezing threshold) but instead considers one single point in the spectrum, e.g. corresponding to the freezing onset. The ice-nucleating properties of INPs in each of the heterogeneous freezing modes are represented by two parameters, i.e. the active fraction ($f_{\mathrm{act}}$) of INPs representing the portion of aerosol particles that lead to the formation of ice crystals, and the critical supersaturation ratio with respect to ice ($S_c$), at which the freezing process is initiated. In the model simulation performed in this study we consider heterogeneous ice nucleation on mineral dust (DU) via both deposition and immersion freezing, and on aviation black carbon (BCav) and black carbon from surface sources (BC) via deposition freezing. The ice-nucleating properties of the different INPs are summarized in Table 1. Further details on the aerosol-cloud coupling are described in Righi et al. (2020, 2021) and Beer et al. (2022).

The EMAC setup applied here is in large parts based on the setup described in Beer et al. (2024). The model simulation is driven by the anthropogenic and open-burning emission data sets developed in support of the CMIP6 project (Gidden et al., 2019; Feng et al., 2020). Emissions for the year 2021 follow the SSP3-7.0 scenario, projecting a radiative forcing of $7.0\,\mathrm{W\,m^{-2}}$ in the year 2100 (O'Neill et al., 2017; Fujimori et al., 2017).

Note that in this study, BCav particles are assumed to act as efficient INPs with $S_c = 1.2$ (see Table 1). However, a recent laboratory study by Testa et al. (2024) analysed black carbon emitted from commercial aircraft engines and determined that BCav has potentially a low ice nucleation efficiency. This suggests that the assumed $S_c$ value is likely an overestimation.





Variations in the ice nucleation ability of BCav not only affect the number concentration of newly formed ice crystals and the
frequency of BCav-induced nucleation events, but also influence heterogeneous freezing on DU and BC, as well as homogeneous nucleation, due to the competition for available water vapor. The decision to assign a high $S_c$ value to BCav was made to simplify the interpretation of the results, making their effect more prominent, and to assess an upper limit for identifying and discussing its potential impact on cirrus formation. For more details about the uncertainty related to this assumption see Righi et al. (2021).

## 3.2 Methodology to enable comparison between observational and model data

In order to ensure the comparability with the aircraft measurements, we use the diagnostic submodel S4D (Sampling in 4 Dimensions; Jöckel et al., 2010) of EMAC to extract model output along the aircraft trajectories of the flights conducted during the CIRRUS-HL campaign at every model time step (i.e., every 15 minutes). To match the model time step, we also average the observed aerosol and ice cloud properties over 15 minutes (an overview of the data before averaging is provided in Sect. 4.1). To improve statistical robustness and account for flight path variations not captured by the model time resolution (e.g., short-term climbs or descents followed by further altitude changes, see Fig. 1 in Jurkat-Witschas et al., 2025), we include model output from $\pm 1$ pressure level around the flight path pressure level. The results obtained using this methodology are presented in Sect. 4.2.

Using the new high-volume flow aerosol particle filter sampler (HERA), integrated into the HALO aircraft to sample INPs (Grawe et al., 2023), we gained insights into INP number and type in the mixed-phase temperature regime. However, these in situ measurements were not suitable for the cirrus regime. Similar to other state-of-the-art INP samplers, HERA faces limitations in providing reliable measurements below $-30\,^\circ$C and does not reach the colder temperatures typically observed in cirrus ($< -38\,^\circ$C). Moreover, for this specific investigation, sampling at low temperatures alone is insufficient, as the focus is on characterizing the influence of INPs along the backward trajectories of the air masses. Therefore, Lagrangian sampling of INPs and clouds would be required to track cloud evolution over time with in situ measurements which represents a technical and operational challenge.

Due to these limitations, we use model data to analyse aerosol effects on cirrus formation and microphysical properties observed during CIRRUS-HL. By using global simulations of INP concentrations and newly formed ice crystals, we can obtain data across different latitudes, longitudes, pressure levels, and times (considering the respective resolution in the model) and combine this information with backward trajectories of the sampled air masses (the results are shown in Sect. 4.3). For this analysis we use a model output interval of 1 hour to align with the time resolution of the backward trajectory data. 10-day backward trajectories were calculated from the flight tracks using the Lagrangian analysis tool LAGRANTO (Wernli and Davies, 1997; Sprenger and Wernli, 2015) and European Centre for Medium-Range Weather Forecasts (ECMWF) analyses. Cirrus trajectories were considered from their formation point onward and classified as "liquid origin" if liquid water was present along the trajectory and as "in situ origin" otherwise, based on the ECMWF reanalysis data (Wernli et al., 2016; Luebke et al., 2016; Krämer et al., 2016; De La Torre Castro et al., 2023).





## 4 Results

### 4.1 Overview of aerosol and ice particle observations during CIRRUS-HL

In this section, we present 1 Hz aerosol and cirrus particle measurements performed during the CIRRUS-HL campaign to analyse their properties at high and mid-latitudes. This overview also serves as a detailed reference for the subsequent analysis and evaluation with the model data described in Sect. 3. Figure 1 displays vertical profiles of three integrated aerosol particle size modes, while Figure 2 presents cirrus microphysical properties as a function of the ambient pressure. The aerosol measurements for particles with $D > 12$ nm obtained from the CPCs consist of 375352 1-Hz samples, and measurements from the OPC total 431907 samples. Cirrus measurements, constrained by the temperature threshold of $-38°$C, include 77087 samples. Both aerosol and cirrus particle data are grouped into mid-latitudes (ML) and high latitudes (HL), using the threshold of $60°$N, as previously applied in De La Torre Castro et al. (2023). High-latitude measurements were less frequent, accounting for $28\%$ of all aerosol samples and $36\%$ of cirrus particle data, respectively.

Given that the primary objective of the campaign was to investigate cirrus clouds, most aerosol measurements in Fig. 1 were performed in the upper troposphere, at altitudes above the 400 hPa pressure level. At lower altitudes, particularly at high latitudes, the median values are associated with larger uncertainties due to limited data availability. At low pressure levels below 500 hPa, the integrated aerosol concentrations for the three size modes do not differ much between high and mid-latitudes. However, a consistent trend emerges at pressure levels above 500 hPa: aerosol concentrations across all three size modes are higher at mid-latitudes than at high latitudes.

The mid-latitude profile of the aerosol concentration of particles with $D > 12$ nm ($N_{\text{aer, } D>12 \text{ nm}}$) aligns with previous studies during the spring and summer seasons, when wildfire activity is typically higher (Norgren et al., 2024). Our aerosol observations at high latitudes are in good agreement with the measurements from Norgren et al. (2024) conducted near Kiruna, Sweden. In addition, our results are also consistent with the tropospheric profiles from the INCA campaign in the Northern Hemisphere (Minikin et al., 2003) and the LACE 98 campaign (Petzold et al., 2002). In contrast to our study, Borrmann et al. (2010) reported lower aerosol concentrations in the tropical upper troposphere. Vertical profiles from Brock et al. (2021) in the remote upper troposphere show an order of magnitude higher accumulation mode particle concentrations compared to our measurements. This difference can be attributed to their lower cut-off diameter of 80 nm for the accumulation mode particles, which results in higher reported concentrations.

Statistically, cirrus clouds at mid-latitudes were observed at higher altitudes than those at high latitudes, as shown in Fig. 2 (d). However, the altitudes observed relative to the tropopause were similar for both latitude ranges. At pressure levels below 225 hPa, high-latitude data is sparse, requiring careful interpretation of the median values of cirrus properties.

Figure 2 (a) shows that mid-latitude (ML) cirrus contain approximately one order of magnitude more ice crystals than high-latitude (HL) cirrus (De La Torre Castro et al., 2023) along the whole pressure profile. The ice water content is slightly higher in ML cirrus at lower altitudes, but overall the difference in $IWC$ between high and mid-latitudes is not pronounced. This is because the higher number concentration at mid-latitudes is compensated by the presence of fewer but larger ice particles at high latitudes, leading to similar $IWC$. $RH_{\text{ice}}$ is generally higher by $9\%$ in median in HL cirrus than in ML cirrus across



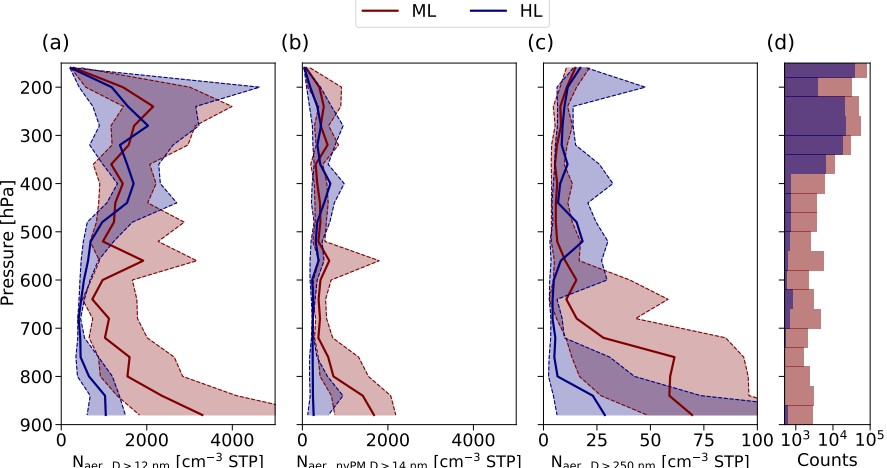

**Figure 1.** Vertical profiles of background aerosol concentrations at high (HL) and mid-latitudes (ML) during CIRRUS-HL for different size ranges: (a) diameters $> 12$ nm measured with the CPCs, (b) diameters $> 14$ nm of non-volatile particle matter (nvPM) measured with the CPCs with thermodenuder, and (c) diameters $> 250$ nm measured with the OPCs. Median profiles are indicated with solid lines, the areas between the $25^{\text{th}}$ and $75^{\text{th}}$ percentiles are filled and indicated within dashed lines. (d) Number of observations per pressure bin of 40 hPa each.

all pressure levels. Gierens et al. (2020) observed an increase in the frequency of ice supersaturation with latitude in the Arctic, while Dekoutsidis et al. (2024) linked high Arctic supersaturations to warm air intrusions. In addition, a relaxation of the relative humidity towards saturation values is observed in ML cirrus at the highest measured altitudes, which is likely influenced by contrail and contrail cirrus (Kaufmann et al., 2014; Schumann et al., 2015).

The central finding of this section is that the difference in aerosol concentrations between high and mid-latitudes does not directly correspond to the observed difference in ice crystal number concentration. While the difference in median aerosol concentrations varies by factors of 2 to 4 at the lower altitudes and converges at higher altitudes, ice crystal number concentrations in cirrus differ by an order of magnitude between high and mid-latitudes. That being said, it is important to note that INPs represent only a small fraction of total aerosol number concentration. Previous studies have used the number of aerosol

particles with diameters larger than 500 nm ($N_{\text{aer}}$, $D > 500$ nm) as a proxy for INP concentrations in the cirrus temperature regime (DeMott et al., 2010; Patnaude and Diao, 2020; Ngo et al., 2025). The median $N_{\text{aer}}$, $D > 500$ nm in our measurements is generally below $0.5$ cm$^{-3}$ at cirrus altitudes (see Fig. S1 in the Supplement), which represents an upper bound of the ice crystal number concentration (assuming heterogeneous nucleation). Similar to $N_{\text{aer}}$, $D > 250$ nm, no clear differences are noticeable between high- and mid-latitude median aerosol concentrations in the upper troposphere, with a maximum median

difference of a factor of 3 at altitudes close to the ground.

Based on these observations, we conclude the following: while liquid origin cirrus clouds form at lower altitudes, where aerosol (and potentially INP) concentrations differ more distinctly between high and mid-latitudes, this alone cannot account




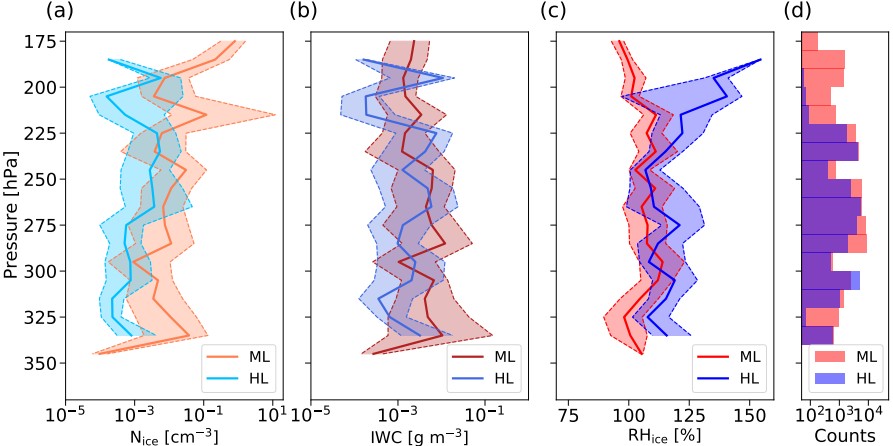

**Figure 2.** Vertical pressure profiles of cirrus microphysical properties and in-cloud relative humidity over ice ($RH_{ice}$) at high and mid-latitudes during CIRRUS-HL: (a) ice particle number concentration ($N_{ice}$), (b) ice water content ($IWC$), and (c) in-cloud $RH_{ice}$. Median profiles are indicated with solid lines, the areas between the $25^{th}$ and $75^{th}$ percentiles are shaded and indicated within dashed lines. (d) Number of observations per pressure bin of 10 hPa each.

for the factor of 10 difference in ice crystal numbers. The substantial difference in ice crystal number concentration between high and mid-latitude is probably not driven by the total number of INPs available at the measurement location. Therefore, we investigate in Sect. 4.3 the properties and distribution of available INP types for each latitude region along backward trajectories and the number concentrations of ice crystals formed from these mechanisms relying on simulations from EMAC.

## 4.2 Observations and model intercomparison

This section presents the results from the direct comparison of in situ measurements of ice particles and aerosols from the CIRRUS-HL campaign with EMAC S4D diagnostic output along the flight tracks, as described in Sect. 3.2. This study primarily serves as a descriptive and qualitative intercomparison between in situ observations and the EMAC model rather than a comprehensive evaluation of model performance. A more thorough assessment would require additional measurements across different regions and seasons. However, the CIRRUS-HL campaign covered a diverse range of meteorological conditions, including remote areas and regions influenced by significant anthropogenic emissions from ground sources and aviation, providing a comprehensive data set for comparison. In addition, the following analyses provide valuable context for the analysis in Sect. 4.3, where aerosol-cirrus interactions observed during the CIRRUS-HL campaign are further discussed using model data.

Performing a direct, one-to-one comparison between observations and model output is challenging mainly due to differences in spatial resolution and an inherent sampling bias in the observations. As explained in Sect. 3.2, to enhance comparability with the model grid-box averages, the observational data were averaged over the model's 15-minute time resolution. Additionally, only ice crystals with diameters smaller than 200 μm are considered in the calculation of cirrus microphysical properties to





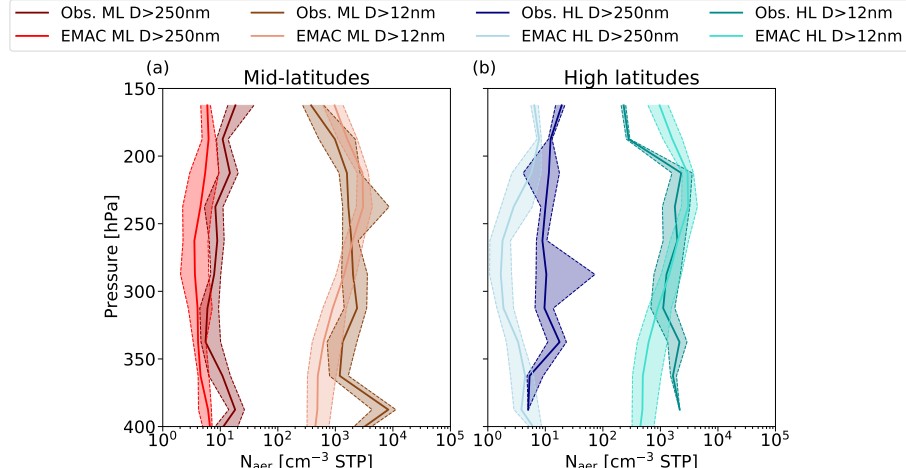

**Figure 3.** Vertical profiles of background aerosol concentrations for two size ranges between the observations and the EMAC model for (a) mid-latitudes and (b) high latitudes. Median profiles are indicated with solid lines, the areas between the $25^{\text{th}}$ and $75^{\text{th}}$ percentiles are filled and indicated within dashed lines.

ensure consistency with the model, where ice crystals larger than $200\,\mu\text{m}$ are transferred to snow crystals, which are assumed to be removed within one model time step by precipitation, melting, or sublimation (Righi et al., 2020).

The analysis in Fig. 3 is constrained to higher altitudes (above the $400\,\text{hPa}$ pressure level), where data density is highest (as shown in Fig. 1), allowing for the application of 15-minute averages. In Fig. 3, darker colors represent profiles from the observations, while lighter colors indicate the model output. Overall, the model mostly shows reasonable agreement with the observations for both size ranges and latitude groups. The cut-off diameter of $12\,\text{nm}$ in the model was selected to match the estimated cut-off from the measurements. Variations in the cut-off diameter of $\pm 2\,\text{nm}$ result in maximum differences of $\pm 10\%$ in $N_{\text{aer},\ D>12\pm2\,\text{nm}}$ in the model. A particularly good alignment is observed in $N_{\text{aer},\ D>12\,\text{nm}}$ at both mid- and high latitudes. However, the model exhibits a significant underestimation for $D > 250\,\text{nm}$ at high latitudes between 350 and 200 hPa, with a maximum discrepancy of one order of magnitude at $300\,\text{hPa}$.

The magnitude of these differences across the pressure range is examined in more detail in Fig. 4 (a) and (b), where it is represented as the order of magnitude of the ratio between observed and modelled medians for mid- and high latitudes in both size ranges. The vertical range in Fig. 4 is extended down to $< 500\,\text{hPa}$ to illustrate the trend at lower altitudes. Additionally, panel (c) compares temperature profiles from both observations and the model, showing a reasonable alignment.

Further examining the trend in $N_{\text{aer},\ D>12\,\text{nm}}$ the difference between observed and modelled values varies with altitude. At the highest altitudes, the model slightly overestimates aerosol concentrations, followed by close agreement in the upper troposphere and increasing discrepancies at lower altitudes. The model's overestimation at high altitudes might result from an overestimation of the upward transport of particles or of the particle nucleation rate (Kaiser et al., 2019). An underestimation of smaller particles in the lower troposphere has been reported in previous studies (Kaiser et al., 2019), likely due to an





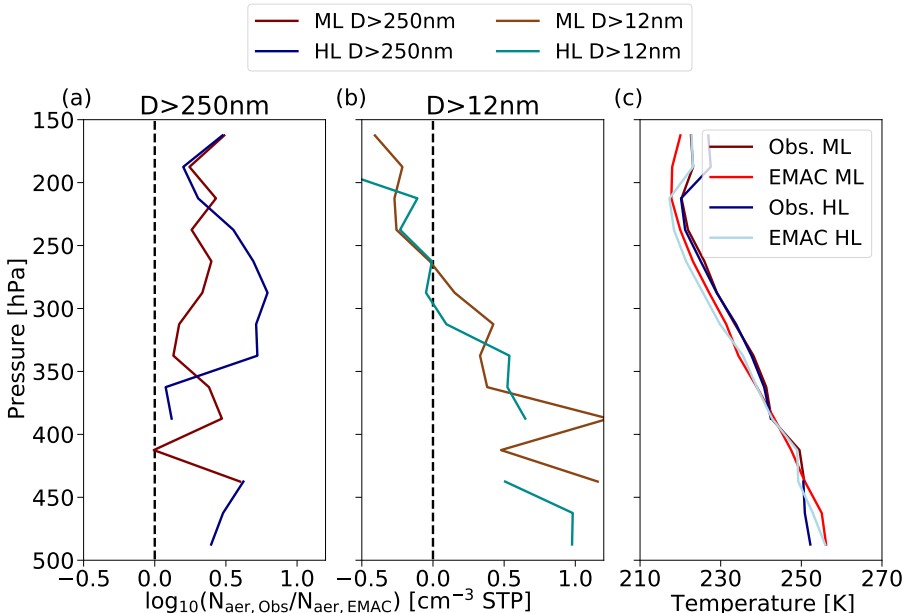

**Figure 4.** Comparison observations and model represented as the ratio of median number concentrations in observations to median number concentrations in the model for (a) $D > 250$ nm, and (b) $D > 12$ nm. (c) Vertical profiles of temperature from observations and model.

underrepresentation of natural aerosol sources and new particle formation from precursor gases in the model. Near the surface (not shown here), the difference becomes more pronounced at mid-latitudes, which could be attributed to a higher density of emission sources compared to high latitudes, which are not fully represented in the emissions data set.

     Ice cloud properties from observations and model as a function of temperature are compared in Fig. 5. The overall medians (panels a, d, and g) are dominated by mid-latitude data points due to lower data coverage at high latitudes. At high latitudes,

there is more variability and fewer data points in both observations and the model, leading to higher uncertainty. However, a general alignment between model and observations is observed in the three cloud properties, indicating lower ice crystal number concentration, lower ice water content, and higher relative humidity over ice at high latitudes compared to mid-latitudes. Climatological averages of these properties for June and July over the period 2014-2021 confirm this trend (see Fig. S2 in the Supplement) and also serve as reference to contextualize the specific CIRRUS-HL episode. However, both model outputs

should not be directly compared due to the different methodological approaches applied.

     In general, the observed ice properties at mid-latitudes in the cold regime ($< 220$ K) are mostly well captured in the model. At the lowest temperature of the measurements ($\approx 212$ K), the model predicts $N_{ice}$ and $RH_{ice}$ comparable to the observations but overestimates $IWC$. Since this high bias in $IWC$ is not reflected in $N_{ice}$, it suggests that the model assumes larger ice crystal sizes than observed, resulting in more ice mass per particle. High $N_{ice}$ at mid-latitudes for this temperature regime

are potentially connected to contrail formation dominating the 15-min averages. De La Torre Castro et al. (2023) indicated that identifiable aged contrails and contrail cirrus appeared in the dataset as outliers, but their influence was likely greater





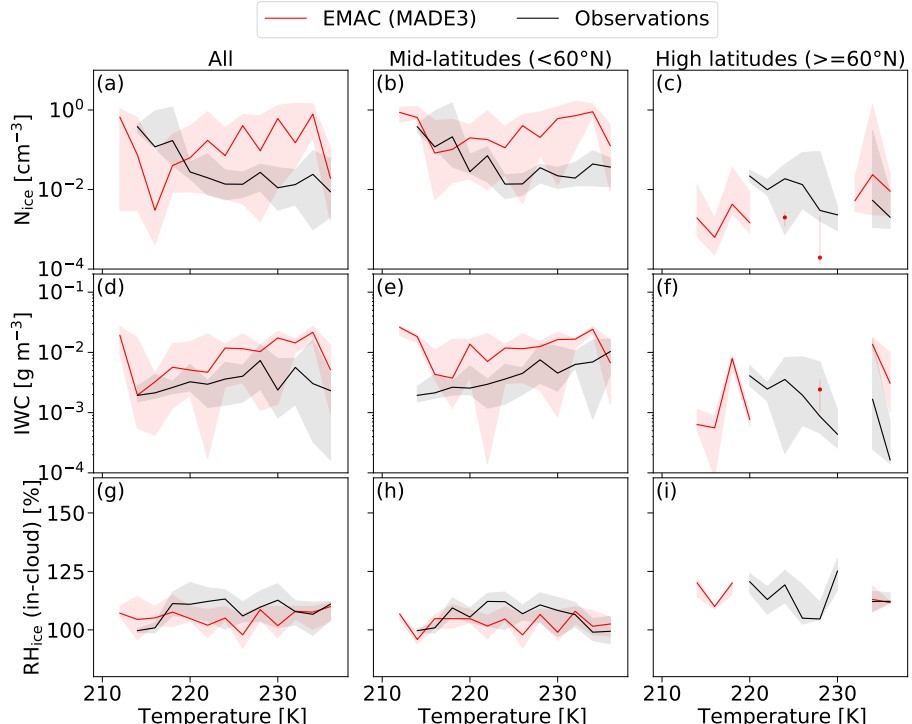

**Figure 5.** Comparison of temperature profiles of microphysical properties of ice clouds between the observations (black) and the EMAC model (red) during the CIRRUS-HL campaign: (a), (b), and (c) ice crystal number concentration ($N_{ice}$); (d), (e), and (f) ice water content ($IWC$); (g), (h), and (i) in-cloud relative humidity with respect to ice ($RH_{ice}$). The left column (a, d, g) represent all data, while the center (b, e, h) and right column (c, f, i) show only mid- and high-latitude data points, respectively. Solid lines indicate median values per 2 K temperature bin and shaded areas extend from $25^{th}$ and $75^{th}$ percentiles. Discontinuous lines appear in the high-latitude panels due to insufficient statistics in those temperature bins.

than assumed, since the conservative selection criteria only accounted for cases clearly distinguishable from natural cirrus and the significant overlap in their microphysical properties made further separation difficult. However, this mechanism is not represented in the EMAC simulations, as the current microphysical scheme does not incorporate the physical processes associated with contrail ice nucleation. As a result, the comparability between model and observations in this temperature range is inherently limited.

For warmer cirrus ($T > 220$ K), the model consistently overestimates both $N_{ice}$ and $IWC$. This discrepancy is particularly evident in $N_{ice}$, where the model exceeds observations by one to two orders of magnitude. A similar bias in this temperature range has been reported in previous studies, potentially linked to an underestimation of ice crystal sizes in the model, which translates into artificially high number concentrations (Bacer et al., 2018; Righi et al., 2020). Specifically, Righi et al. (2020) observed this bias when comparing model results with climatology data sets from Krämer et al. (2009) and Krämer et al. (2016), and with the observations from the ML-CIRRUS campaign (Voigt et al., 2017). Figure S3 in the Supplement compares the



ML-CIRRUS and CIRRUS-HL median number concentrations from both observations and model output from climatological averages over the corresponding periods, revealing similar profiles between campaigns.

It is important to note that we compare the model output from ML-CIRRUS (Righi et al., 2020) only with the corresponding climatological average for CIRRUS-HL, and not with the S4D model output used in this study. As in Righi et al. (2020), the climatological model data were averaged vertically and horizontally over the relevant campaign region, potentially leading to a smoothing effect on the medians, whereas the S4D output along the flight tracks allows for a more direct one-to-one comparison with the observations. It should also be noted that these campaigns occurred under different meteorological conditions and

seasons. The CIRRUS-HL campaign meteorology was influenced by strong convective activity during summer (Tomsche et al., 2024; Jurkat-Witschas et al., 2025). Previous studies by Gasparini et al. (2018); Muench and Lohmann (2020) have reported an overestimation of ice crystal number concentration in the ECHAM-HAM model compared to observations, particularly for ice crystals detrained from convective events. This overestimation may be due to inadequately represented microphysical processes, such as collision-coalescence, aggregation, or sedimentation, which reduce ice crystal number concentrations

while increasing particle size during convective outflow (Jensen et al., 2009). We analyse the influence of detrainment from convection on the $N_{\mathrm{ice}}$ in Sect. 4.4 in more detail.

## 4.3   Aerosol-cirrus interaction

De La Torre Castro et al. (2023) showed that high-latitude cirrus exhibit on average lower number concentrations but larger ice crystal sizes compared to mid-latitude cirrus, along with higher supersaturation. Here, we investigate whether these differences

in cirrus microphysical properties could result from a reduced availability of INPs at higher latitudes. To gain further insights into the life cycle of the observed cirrus and their interaction with aerosol, we use global simulations of INP concentrations from the EMAC model (with hourly output), as described in Sect. 3.2, after having validated in Sect. 4.2 the model's ability to reproduce the CIRRUS-HL observations with reasonable accuracy. We focus on heterogeneous nucleation processes as the main driver of the observed cirrus properties, because the analysis of both measured vertical updrafts and derived vertical up-

drafts along the backward trajectories (interpolated from ERA5) indicated a negligible influence of homogeneous nucleation in our data set. The hourly model output along the backward trajectories further confirms that only a few homogeneous nucleation events occurred in the analysed time period in the model (see Fig. S4).

    Figure 6 illustrates the methodology for determining INP concentrations (DU, BCav, and BC) and newly formed ice crystal concentrations ($N_{\mathrm{i,\ DU}}$, $N_{\mathrm{i,\ BCav}}$, and $N_{\mathrm{i,\ BC}}$) along backward trajectories for two cirrus types: M-M (formed and measured at mid-latitudes) and H-H (formed and measured at high latitudes). De La Torre Castro et al. (2023) found that high-latitude cirrus

could be divided into two groups depending on their latitude at formation. When air masses originated from mid-latitudes, the resulting cirrus (M-H) exhibited higher number concentrations and smaller ice crystals. Given that we aim to explore influences along the cirrus life cycle, we further distinguish cirrus by their latitude of origin and, for simplicity, consider only "pure" mid-latitude or high-latitude cirrus (M-M and H-H).

In Figs. 6 (b), (d) and (f), we present absolute frequency distributions of all INP-induced nucleation events occurring at each model time step (every hour) along the backward trajectories of M-M (orange) and H-H (blue) cirrus. The absolute frequency



values serve as a reference for comparison, as they depend on the time step (shorter intervals result in higher frequencies) and the length of the trajectories. M-M and H-H cirrus have similar median lifetimes of approximately 16 and 22 hours, respectively.

Examining the frequency of events for each INP type, it becomes evident that ice crystals predominantly form on mineral dust particles due to their strong freezing potential and higher concentration at lower latitudes (Möhler et al., 2006; Froyd et al., 2022), as reflected by the broad distributions in Fig. 6 (b). Freezing on DU can initiate at lower supersaturations compared to BCav or BC. However, under suitable conditions, freezing on BCav and BC can also occur. As shown in Fig. 6 (a), mineral dust availability is higher at mid-latitudes, making it likely that cirrus forming at these latitudes are more influenced by these

nucleating particles. Figure 6 (b) further highlights a clear difference between the distributions of ice crystals nucleated by DU in M-M and H-H cirrus, with a pronounced second mode at approximately $0.1 \ \mathrm{cm}^{-3}$ for M-M cirrus. Ngo et al. (2025) evaluate the influence of various parameters on cirrus properties and also find that $RH_{\mathrm{ice}}$ plays a more decisive role than the total concentrations of both larger aerosol particles ($> 500 \ \mathrm{nm}$) and smaller ones ($> 100 \ \mathrm{nm}$).

    Interestingly, the influence of BC from ground sources is minimal in H-H cirrus in terms of absolute frequency of events,

whereas BCav appears to have a comparable effect in both M-M and H-H cirrus. BCav exhibits a stronger influence in M-M cirrus, where the median $N_{\mathrm{ice}}$ is an order of magnitude higher than in H-H cirrus. It is important to note that the ice crystal number concentrations and the frequency of events are very likely overestimated, as suggested by recent findings by Testa et al. (2024). However, these results are still useful for assessing the potential role of BCav as an INP. If BCav was an efficient INP, it would have a noticeable impact on cirrus formed at high latitudes, while BC from other sources than aviation seems less

relevant. This is because H-H cirrus are mainly of in situ origin ($\approx 90\%$, De La Torre Castro et al., 2023), forming at high altitudes and remaining within that range throughout their trajectories. While black carbon concentrations from ground sources decrease with altitude, aviation emissions create a distinct concentration peak of BCav at cruise altitudes (Righi et al., 2021; Beer et al., 2022), precisely where in situ cirrus evolve.

    It should be noted that while the role of homogeneous nucleation in ice crystal formation during CIRRUS-HL was neglected,

we cannot exclude a contribution from contrails to the overall ice crystal number concentration, as discussed in Sect. 4.2.

## 4.4   Model refinement based on observational insights

In Sect. 4.2, we identified an overestimation of ice crystal number concentration by the model at mid-latitudes at warm temperatures (above 220 K). This is likely connected to the higher frequency of convective events at mid-latitudes, while this influence was minimal at high latitudes. In this section, we examine the role of convection in the model overestimation of $N_{\mathrm{ice}}$

in more detail and focus our analysis on the temperature region above 220 K.

    In EMAC, the ice crystal number concentration from convective detrainment is calculated based on the radius parameterization for convective ice crystals of Boudala et al. (2002). In Fig. 7, we compare this approach with two alternative ice crystal radius parametrizations derived from CIRRUS-HL measurements during convective events over the Alps (flights F12 and F15, excluded from the main analysis in the previous sections). Based on targeted EMAC simulations, we then assess how these

new fits affect the model output compared to the reference case using the Boudala et al. (2002) fit (denominated as "ref").



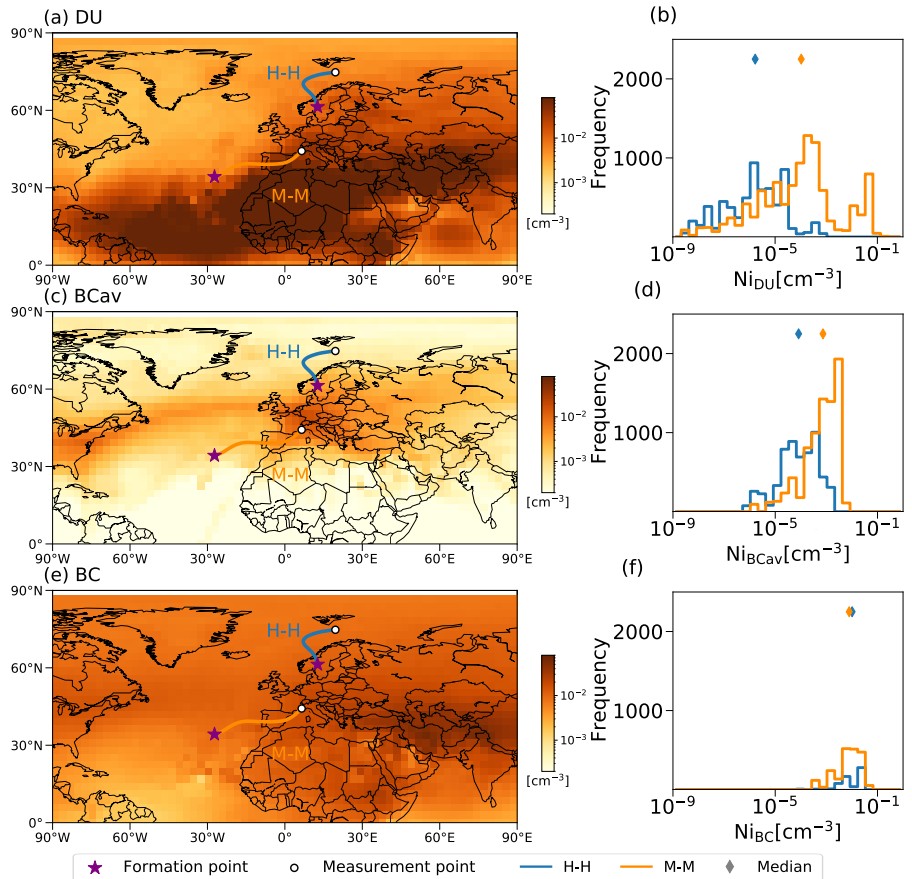

**Figure 6.** Geographical distribution of the simulated number concentration of INPs inside cirrus clouds considering the multi-year average over the June/July months of the years 2014-2021 and over all vertical levels. The number concentrations of potential INPs are weighted with ice-active fractions at ice supersaturations of $S_i = 1.4$ (see Table 1). This procedure follows the approach described in Beer et al. (2022). Shown are (a) dust (DU), (c) black carbon from aviation (BCav), and (e) black carbon from other ground sources (BC). Each panel (a, c, e) also includes an example backward trajectory for each cirrus type, with a purple star indicating the formation point and a white-black dot representing the measurement location. Absolute frequency distribution of newly formed ice crystals from (b) DU, (d) BCav, and (f) BC along all backward trajectories of cirrus formed and measured at high latitudes (H-H, blue), and cirrus formed and measured at mid-latitudes (M-M, orange). The medians of both groups are indicated with diamonds in the upper part of the plots. The frequency distributions of newly formed ice crystals along trajectories correspond to the hourly output for June-July 2021.

Figure 7 (a) shows the convective ice crystal radius as a function of temperature. The reference fit from Boudala et al. (2002) is shown with a solid red line, and the two CIRRUS-HL-derived fits are represented by dashed and dotted red lines. The CIRRUS-HL fits are only valid for the temperature regime > 220 K and smooth transitions to the Boudala et al. (2002) fit are implemented at this temperature threshold to account for the colder temperature regime (using a sigmoid logistic function with a steepness parameter of $k = 1$). These new parameterizations are based on the mean volume diameter of the ice crystal



populations per sample period, as defined in Schumann et al. (2011). The mean volume diameter is calculated after using two different methods to estimate the diameter of individual ice particles: (fit1) the maximum dimension of each ice crystal, calculated as the minimum enclosing circle; and (fit2) the area-equivalent diameter, derived from the projected surface area of the ice crystals (Wu and McFarquhar, 2016). The resulting particle size distributions are used to calculate mean volume

diameters, which are then binned and fitted as a function of temperature to produce the final parameterizations. Fit1 consistently yields larger radii than fit2, and both considerably exceed the values from the Boudala et al. (2002) fit. This could be due to the fact that data sets from the early 2000s are likely affected by the ice crystal shattering phenomenon described in many studies, e.g., Field et al. (2003, 2006); McFarquhar et al. (2007); Korolev et al. (2011, 2013), which artificially increases the number of small ice crystals, leading to an underestimation of the mean volume diameter.

Figure 7 (b) shows the absolute frequency distribution of newly formed ice crystals from convective clouds ($N_{\text{i, CV}}$), computed along backward trajectories, analogous to the analysis in Fig. 6 (b), (d), and (f) for DU, BC, and BCav, respectively. At high latitudes, the influence of convective detrainment is negligible. However, at mid-latitudes, the median $N_{\text{i, CV}}$ for the reference case is around $1\ \text{cm}^{-3}$, significantly higher than the concentrations of newly formed ice crystals from DU, BC, and BCav. Both fit1 and fit2 reduce the median $N_{\text{i, CV}}$ by one to two orders of magnitude, with fit1 having a stronger effect due to

its larger estimated radii. This reduction results from distributing the same ice mass among fewer, but larger crystals.

Finally, Fig. 7 (c) illustrates the impact of the different radius fits on the comparison between modelled and observed total ice crystal number concentrations as a function of temperature for mid-latitude cirrus, as in Fig. 5 (b). Both alternative fits improve the model estimates compared to the reference case at temperatures above 220 K. Although fit1 seems to match the observations more closely, it may lead to an overcorrection of $N_{\text{ice}}$. In contrast, the area-equivalent diameter method used for

calculating the mean volume diameter leading to fit2 may provide a more reliable estimate of the actual ice crystal volume, particularly in the presence of complex ice shapes, where the minimum-enclosing-circle method would tend to overestimate it. In addition to the CIRRUS-HL specific comparison shown in Fig. 7 (c), we also compare the simulated ice crystal number concentrations to measurement data from a climatology derived from various flight campaigns compiled by Krämer et al. (2020) in a similar manner as presented in Righi et al. (2020) (see Fig. S5 in the Supplement). This comparison also shows an

improved agreement of the simulated results with the observations for temperatures above 220 K when using the alternative fits. Hence, this model refinement represents a valuable way to improve the model representation of cirrus clouds also for other regions and cloud conditions.

These findings allow us to close the loop in our study by improving the model based on observational data. We demonstrate the strong sensitivity of simulated ice crystal number concentrations in EMAC to the model representation of convective

ice crystals and show that applying fits based on CIRRUS-HL measurements substantially improves model accuracy in the temperature range above 220 K.




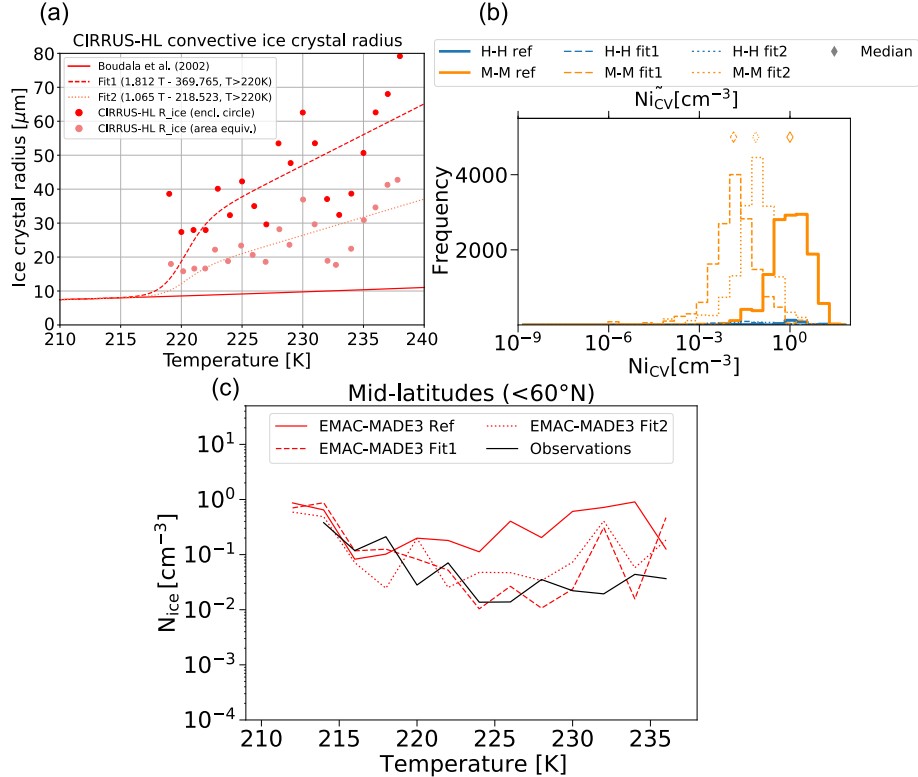

**Figure 7.** (a) Parametrizations for the radius of detrained ice crystals applied in EMAC. The solid line ("ref") represents the reference parametrization from Boudala et al. (2002) applied otherwise in this work and the dashed ("fit1") and dotted lines ("fit2") correspond to fits derived from CIRRUS-HL flights in convection (F12 and F15) for $T > 220$ K and a smooth transition to the reference parametrization for $T < 220 \ K$. (b) Absolute frequency distribution of newly formed ice crystal number concentrations detrained from convection ($N_{\mathrm{i, \ CV}}$) along all backward trajectories of cirrus formed and measured at high latitudes (H-H, blue), and cirrus formed and measured at mid-latitudes (M-M, orange). A linear vertical scale has been used to focus on the M-M category. The median $N_{\mathrm{ice}}$ of M-M cirrus are indicated with diamonds in the upper part of the plot. (c) Same as 5 (b) but including additional EMAC simulations applying fit 1 (dashed line) and fit 2 (dotted line). The lines representing the medians and percentiles are omitted for clarity.

## 5 Summary and conclusions

In this study, we present a combined approach to evaluate and advance our understanding of aerosol-cirrus interactions by analysing in situ ice and aerosol observations from the CIRRUS-HL campaign together with simulations from the global aerosol-chemistry-climate model EMAC with the aerosol submodel MADE3. The analysis of in situ measurements revealed an order of magnitude difference in ice number concentrations between mid-latitude and high-latitude cirrus. However, aerosol concentration integrated across various size ranges did not show a correspondingly large difference between the two regions. In order to evaluate whether the observed patterns could be reproduced by the model, we developed a targeted strategy for accurate intercomparison of aerosol and ice properties, extracting the model output along the flight tracks at 15-minute intervals.



Moreover, we investigated the contributions of different heterogeneous freezing mechanisms to the observed cirrus properties using backward trajectories in combination with EMAC model data. This enabled the derivation of ice-nucleating particle concentrations and number concentrations of newly formed ice crystal along the cirrus life cycle and allowed us to assess their respective influences at high and mid-latitudes. Finally, to investigate a model overestimation of the ice crystal number concentration at mid-latitudes, we tested alternative parameterizations for convective ice crystal sizes derived from CIRRUS-HL measurements and showed that they significantly reduce the modelled ice number concentration and improve agreement with observations. The main conclusions of this study are:

- The observed one-order-of-magnitude higher ice crystal number concentration in mid-latitude cirrus compared to high-latitude cirrus could not be explained by differences in total aerosol concentrations. In particular, the number of particles larger than $500 \, \mathrm{nm}$ (a commonly used proxy for estimations of INP concentrations) does not show a significant difference between the two regions. We attribute the difference in cirrus microphysical properties primarily to the varying influence of specific INP types with distinct freezing efficiencies and its interaction with the ambient supersaturation within the mid- and high-latitude regions, rather than to differences in total INP concentrations. The analysis of the simulated freezing mechanisms along backward trajectories from the measurement locations and the observed updrafts suggests a negligible contribution of homogeneous nucleation in the CIRRUS-HL measurements both at high and mid-latitudes. However, ice crystals detrained from convective systems were of particular importance at mid-latitudes during the campaign period leading to high ice crystal number concentrations. A further contributing factor is the more frequent formation of contrails at mid-latitudes due to denser aviation activity, which increases the ice number concentration and reduces the effective particle diameter.

- Further analyses of the EMAC output along the backward trajectories highlight a prominent role of mineral dust in mid-latitude cirrus formation, leading more frequently to higher ice crystal number concentrations compared to cirrus at high-latitude. According to the model simulations, aviation black carbon was found to have the potential to significantly contribute to high-latitude cirrus formation. This is consistent with the largest fraction of those cirrus being of in situ origin, which have air mass trajectories remaining at cruise altitudes, with the highest concentration of aviation emissions. However, the magnitude of this contribution is subject to the assumed very efficient freezing ability of aviation black carbon particles, which is used to investigate the sensitivity of natural cirrus clouds to modifications caused by these particles when they act as effective ice nuclei.

- The EMAC model mostly shows reasonably good agreement with aerosol observations from CIRRUS-HL. However, a significant underestimation of aerosol number concentration is identified near $300 \, \mathrm{hPa}$ for particles with diameters larger than $250 \, \mathrm{nm}$ at high latitudes, and a slight overestimation is present at higher altitudes for particles larger than $12 \, \mathrm{nm}$ in both cirrus types. Toward lower altitudes, the model tends to underestimate the smaller particle size mode, likely due to underestimated emissions from ground sources and mechanisms of new particle formation from precursor gases which are not yet fully represented in the model.



– At mid-latitudes, the model captures ice properties well in the cold temperature regime ($T < 220$ K), but this agreement is likely coincidental and subject to higher uncertainty due to the assumed ice crystal size in the model and the influence of contrail cirrus on the observations. However, for warmer temperatures ($T > 220$ K), it overestimates the median ice crystal number concentration by about one order of magnitude, consistent with previous findings (Righi et al., 2020). The model and observations appear to agree at high latitudes, but large uncertainties exist due to fewer data points. We mainly attribute the model overestimation of ice crystal number concentration at mid-latitudes at warmer temperatures to an overestimation of the number of ice crystals detrained from convective clouds. We present two alternative parametrizations of ice crystal sizes in the convective outflow derived from the CIRRUS-HL measurements during convective events and demonstrate a significant reduction of this model bias in the estimation of ice crystal number concentrations.

In summary, this study demonstrates the importance of investigating aerosol-cirrus interactions to better understand the microphysical properties of cirrus clouds, which largely determine their radiative impact. Accurate measurements of ice-nucleating particles at the low temperatures typical of the cirrus regime remain a critical need to advance our understanding of ice nucleation mechanisms and their role in the climate system. Meanwhile, the integrated use of model data with in situ observations has proven effective in supporting the interpretation of field measurements and improving our understanding of cirrus formation processes.

*Data availability.* Processed data from the CIRRUS-HL campaign are available at https://halo-db.pa.op.dlr.de/ from the HALO database (DLR, 2025). The model setup and the simulation data analysed in this work will be made available with the final version of this manuscript at https://doi.org/10.5281/zenodo.15534818

*Author contributions.* EDLTC, CB, TJW, MR, and JH conceptualized the study. EDLTC conducted the analyses, performed and evaluated ice particle measurements, and wrote most parts of the paper. CB performed the model simulations, wrote the majority of Sect. 3, and contributed to the analyses. DS performed and provided the aerosol in situ measurements. TJW and CV coordinated the CIRRUS-HL mission. TJW, DS, MR, JH, and CV supervised the study and supported the results interpretation. All authors contributed to and commented on the manuscript.

*Competing interests.* The authors declare that they have no conflict of interest.

*Financial support.* This work is supported by the German Research Foundation within SPP-1294 HALO (grant nos. VO1504/6-1, VO1504/7-1 and VO1504/9-1) and TRR 301 (Project-ID 428312742). This study has been supported by the DLR space research programme (project MABAK).



*Acknowledgements.* We are grateful to Heini Wernli (ETH, Switzerland) for providing backward trajectories of air masses, and to Qiang

Li (DLR, Germany) for his comments and suggestions on an earlier version of the manuscript. CV, TJW, DS, and EDLTC would like to express their gratitude to the Flight Experiments Department at the German Aerospace Center (DLR) for their support during the CIRRUS-HL campaign and for providing aircraft reference measurement data (aircraft position, velocity, and ambient temperature) and relative humidity measurements used in this work. The model simulations and data analysis for this work used the resources of the Deutsches Klimarechenzentrum (DKRZ, Germany) granted by its Scientific Steering Committee (WLA) under project ID bd0080 and bb1393. In

addition, data sets provided by CMIP6 via the DKRZ data pool were used. CB, JH, and MR are grateful for the support of the whole MESSy team of developers and maintainers.





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
