# Peer review of "A combined observational and modelling approach to evaluate aerosol-cirrus interactions at high and mid-latitudes"

_EGUsphere, 2025_

## Author Comment (AC1)

We would like to thank both reviewers for their constructive comments and helpful suggestions. They led to interesting discussions and to a higher quality of the revised manuscript. We want to note that during the revision process, a bug in the calculation of the parametrizations in Sect. 4.4 was detected and corrected, leading to an updated version of Fig. 7. We emphasize that the correction did not alter the statements done in this regard. Additionally, similarly as in Fig. 7(c) for ice particle number concentration, we added comparisons for ice water content, ice particle diameter, and relative humidity as a new Fig. S6. All the reviewers' comments (in black) and corresponding responses (in blue) are listed below. Copied text from the manuscript is marked in italics and underlined sentences correspond to changes in the manuscript (see also the submitted manuscript with track-changes).

**Referee #1**

*Recommendation: Major Revision*

While the manuscript presents valuable results, several substantive points require clarification or additional detail before publication. These include conceptual clarifications, methodological considerations related to model calibration and evaluation against the same observational dataset, and further details on instrument characteristics, data processing, and figure interpretation. Some statements in the conclusions and discussion could also be more clearly supported by the data or relevant literature. Addressing these points will strengthen the manuscript and improve its clarity and impact.

Major comments:

- A key aspect that requires clarification is whether secondary ice production (SIP) mechanisms (such as rime-splintering, ice-ice collisions, or fragmentation) were considered as potential contributors to the observed cirrus ice number concentrations, in addition to the role of primary ice-nucleating particles. SIP has been shown to substantially increase ice crystal numbers not only in mixed-phase clouds but also in convective outflows that may feed cirrus anvils (Hallett and Mossop, 1974; Field et al., 2006, 2017; Korolev and Field, 2008).

- Even in the upper troposphere, SIP may influence ice crystal populations in cirrus formed from convective detrainment, which is particularly relevant for mid-latitude cirrus where high Ni values were observed during the CIRRUS-HL campaign. Neglecting SIP could lead to an incomplete attribution of the observed differences between mid- and high-latitude cirrus. I strongly recommend the authors explicitly discuss whether SIP processes were included in their analysis or simulations, and if not, acknowledge this limitation and its potential impact on the results.

Thank you for the comment. We address both comments on the SIP contribution jointly. The model does indeed simulate SIP, i.e. the Hallet-Mossop mechanism (rime splintering) is parametrized according to Levkov et al., 1992. However, uncertainties remain as other types of ice multiplication processes are missing in the model parametrization, e.g. droplet shattering and ice-ice collisional breakup. With regard to the observations, SIP observations were not an objective of the campaign and thus, its effects have not been explicitly considered in this study. We acknowledge a potential influence in our measurements and include a short discussion at the end of Section 4.2:

*Secondary ice production (SIP) has been shown to substantially increase ice crystal number concentrations not only in mixed-phase clouds but also in convective outflows feeding cirrus anvils (e.g., Hallett and Mossop (1974); Field et al. (2016); Korolev et al. (2022)). As explained in Sect. 3.1, the EMAC model represents some SIP mechanisms, subject to certain limitations. However, we did not perform a dedicated analysis of SIP effects in our observations. While we acknowledge a potential influence on cirrus formed from convective detrainment, the events with high particle concentrations analysed in this study were mostly associated with observations of aged contrails and contrail cirrus.*

We also added the following to the text in Section 3.1: We note that the model does simulate secondary ice production (SIP), i.e. the Hallet-Mossop mechanism (rime splintering) is parametrized according to Levkov et al., 1992. However, uncertainties remain as other types of ice multiplication processes are missing in the model parametrization, e.g. droplet shattering and ice-ice collisional breakup, which should be explored in future studies.

- The methodology appears to adjust the model using observational data (e.g., for fit1 and fit2) and then evaluates the agreement with the same dataset. This approach risks overestimating model performance, as the apparent improvement may largely reflect the fact that the model was calibrated to these observations. The authors should clarify this point and, if possible, include an independent validation dataset to assess whether the fits genuinely improve predictive skill rather than just reproducing the observations used for adjustment (linked to Section 4.4, lines 479–481).

As we indicate in this sentence: *In Fig. 7, we compare this approach with two alternative ice crystal radius parametrizations derived from CIRRUS-HL measurements during convective events over the Alps (flights F12 and F15, excluded from the main analysis in the previous sections)"*, the fits are performed with the flights targeted in convective systems F12 and F15, which are explicitly excluded from the general dataset used for the analysis in the other sections and in the evaluation of those parametrizations shown in Section 4.4. Additionally, we compare the results of the simulations with the modified ice crystal radius parametrizations also against an observational climatology derived from various flight campaigns (Krämer et al., 2016, 2020) in Fig. S7 in the supplement, confirming the results obtained from the CIRRUS-HL comparison. As further clarification, we added the following sentence in the caption of Fig. 7(c):

*Flights F12 and F15, which were used for the parametrizations, are excluded from this data set, consistent with the approach in the other analyses of this study.*

Minor comments:

- Lines 34-35: The statement that homogeneous nucleation occurs mainly in the uppermost troposphere and lower stratosphere is inaccurate. It primarily occurs in the cold upper troposphere under high ice supersaturation and is generally rare in the lower stratosphere. Please revise for accuracy.

We thank the reviewer for this note and modified the lines as follows:

*While heterogeneous nucleation dominates in many regions of the globe and across different altitudes in the upper troposphere, especially over the polluted Northern Hemisphere (Cziczo et al., 2013), homogeneous nucleation primarily occurs in the cold upper troposphere under high ice supersaturation.*

- Lines 40-44: The paragraph beginning with "Understanding and accurately representing aerosol-cloud interactions..." is somewhat disconnected from the previous discussion of

nucleation processes and INP uncertainties. Adding a transition sentence linking general nucleation uncertainties to Arctic observations or regional differences in cirrus properties would improve clarity.

We thank the reviewer for this suggestion. We added the following transition:

*These uncertainties become even more pronounced in remote regions such as the Arctic, where observations on aerosol type, size and number, and nucleation pathways are particularly scarce. However, understanding and accurately representing [...]*

- Lines 80-87: The paragraph describing the flights is somewhat ambiguous regarding convective systems. It states that two flights specifically targeted convective systems were excluded from the main analysis (except in Sect. 4.4), but the campaign also experienced generally unstable conditions with thunderstorms. Please clarify whether any other flights encountered convective systems and, if so, explain why they were included in the analysis while the two targeted convective flights were excluded.

We excluded the two flights that explicitly targeted deep convective systems because their primary purpose was to sample active convection and the associated outflow, which introduces local dynamical conditions (e.g., strong turbulence, high updrafts...) not representative of the synoptic-scale cirrus conditions investigated in this study. However, to better understand the context and influences of the campaign period, we note that it was generally characterized by unstable conditions with frequent thunderstorms over parts of Europe. The remaining flights did not actively penetrate in convective systems, and when isolated convective cells were encountered during transfer-back flight legs, these were mostly avoided. This implies that we cannot exclude the potential measurements in convective outflows, but these were random cases and not comparable with the two excluded flights that were specifically targeted to investigate convection. Section 4.4 includes those cases due to their relevance for understanding and improving the model representation of convection.

We modified the mentioned paragraph to enhance its clarity as follows:

*In this study, we focus on the cirrus and aerosol properties observed during the CIRRUS-HL campaign in summer 2021 with the High Altitude and LOng range research aircraft (HALO) (Jurkat-Witschas et al., 2025). The mission included 24 flights under various meteorological conditions, covering a wide region from mid- to high latitudes over Central Europe, the North Atlantic, and the Arctic, reaching up to 76°N. The synoptic situation during the campaign was characterized by unstable conditions with enhanced moisture (Krüger et al., 2022), leading to frequent thunderstorms and hail over Western and Central Europe. While such conditions were present during the campaign, the flights included in our analysis did not intentionally penetrate or sample active convective systems. However, two flights were specifically targeted in convective systems (Kalinka et al., 2023; Jurkat-Witschas et al., 2025) and were excluded from the main analysis (except in Sect. 4.4, where the model representation of convection is discussed). In total, approximately 25 h of in situ ice particle data and 104 h of in situ aerosol data were collected.*

- Lines 98-100: The previous paragraph notes that the CDP has a 1-second time resolution, but the time resolution or sampling frequency of the CIPgs and PIP instruments is not provided. Please include this information to allow proper comparison and assessment of the temporal representativeness of the measurements.

We make here a general comment that refers to this point and the following three. We tried to simplify and summarize the ice particle data description. Given that the same data set has been

*previously explained in detail in prior publications, we tried to avoid unnecessary repetitions and focus more on the results of the present study. We indicated this in lines 90-91: A detailed description and characterization of the instrumentation and data set are available in De La Torre Castro et al. (2023) and De La Torre Castro (2024). Further instrumentation of the mission is described in Jurkat-Witschas et al. (2025).*

*However, since both reviewers made comments in this regard, we made some additions in the manuscript and note them under the corresponding points:*

*Regarding the reviewer's comment on the probe's time resolution, we moved the sentence from CDP to the last paragraph and reformulated as follows:*

*Particle number concentrations from each three instruments were measured at a 1-second time resolution.*

- Lines 98-106: The CIPgs and PIP instruments have overlapping particle size ranges. Please clarify how particle data in the overlap region were handled to ensure consistency between the two instruments.

*We included the ranges for calculating the combined particle size distribution and referred to publications for more details:*

*To obtain a complete particle size distribution, data from the three instruments were combined (CDP: 2 – 37.5 μm; CIPgs: 37.5–247.5 μm; average CIPgs-PIP: 247.5–637.5 μm; PIP: 637.5–6400 μm; further details can be found in De La Torre Castro et al. (2023) and Sect. 4.3 of De La Torre Castro (2024)).*

- Lines 105-106: The manuscript mentions that one-pixel images and other artifacts were excluded. Please specify what types of artifacts were identified and removed from the dataset.

*Here we refer to electronic noise and added the following in the manuscript for clarification:*

*One-pixel images and other artifacts from electronic noise were identified and excluded from the analysis.*

- Lines 107-110: The paragraph introduces the mass-dimension relationship and mentions that ice crystal number concentration (Nice) is derived from the combined dataset. Please clarify that D represents the ice particle diameter obtained from the probe measurements. Additionally, explain how Nice was derived from the combined instruments. Was it calculated directly from particle counts, or were corrections applied to account for overlapping size ranges and differences between probes?

*We made the following additions in those lines:*

*We calculate the IWC with the mass-dimension relationship $m = a \cdot D^b$ (in cgs units) with coefficients a = 0.00528 and b = 2.1, as proposed by Heymsfield et al. (2010), and where D represents the particle diameter measured by the cloud probes. Nice was calculated directly from the sized resolved particle concentrations obtained from the combined distribution.*

- Lines 114-118: The size ranges for the CPC and OPC are provided, but for clarity, please indicate the overall maximum particle diameter measurable by AMETYST.

*We repeated the overall maximum diameter in the general sentence of the measurement system AMETYST (comprising OPCs and CPCs) for clarity:*

*The aerosol measurement system utilized by the German Aerospace Center (DLR) aboard the HALO aircraft is known as the AMETYST (Aerosol MEasuremenT sYSTem) and measures aerosol particles between 12 nm up to approximately 3 μm.*

- Lines 150-153: The authors state that the simulations were performed in nudged mode. Could they discuss the potential impacts or limitations this approach may have on the results?

Indeed, nudging could possibly suppress feedback mechanisms that would occur in a free-running simulation where the meteorology is not influenced by predefined values. However, it has been shown in Beer et al. (2024), that the differences in e.g. ICNC between a nudged and a free-running simulation are small at cirrus altitudes (mostly < 10%). Importantly, nudging is essential for comparisons with observational campaign data to achieve realistic meteorological conditions in the model. In addition, the calculation of wind-driven mineral dust emissions in the model relies on the nudged model set-up to produce reasonable dust emission values (Beer et al., 2020).

- Lines 174-175: Please clarify whether SSP3-7.0 was chosen instead of SSP2-4.5 or other scenarios. Was this selection specifically intended to explore a high-emission or strong-forcing pathway?

We use anthropogenic and open-burning emissions from the CMIP6 project representative of the SSP3-7.0 scenario. In the year 2021, when the campaign tool place, the differences between the SSP scenarios are still very small and the results of the present study would be very similar for emissions representative of e.g. the SSP2-4.5 scenario.

- Lines 22-211: Please clarify whether the global simulations used for the backward-trajectory analysis are the same as the EMAC + MADE3 model runs described earlier, or if a different model setup was used. Additionally, please provide more details on how clouds were classified as "liquid origin" versus "in situ origin" based on the presence of liquid water along the trajectory, such as how much liquid water, at which levels, and based on what threshold.

The reviewer is right, both simulations are actually the same EMAC simulation, in one case we calculate the model output along the flight tracks from the S4D submodel with a time resolution of 15 min, and in the other case, we adjust the output resolution to 1 hour to match the backward trajectories and calculate the model output along the backward trajectories.

We adjusted those lines accordingly:

*For this analysis, we use the same model configuration as in the S4D simulations described above, but without applying the S4D submodel and with an output interval of 1 hour to align with the time resolution of the backward trajectory data.*

Regarding the cirrus classification, this was also explained with more detail previously in De La Torre Castro et al. (2023). We made some adjustments in the paragraph to provide more details:

*Cirrus trajectories were considered from their formation point onward, which was determined as the last time step before IWC = 0 when tracing the trajectory backward. They were classified as "liquid origin" if liquid water was present along the trajectory (LWC > 0) and as "in situ origin" otherwise, based on the ECMWF reanalysis data (Wernli et al., 2016; Luebke et al., 2016; Krämer et al., 2016; De La Torre Castro et al., 2023).*

- Lines 224-225: I would be cautious with the statement that at lower altitudes, particularly at high latitudes, the median values are associated with larger uncertainties due to limited data

availability. Figure 1 shows that this is not consistently the case across all aerosol size ranges. For example, for D > 14 nm the uncertainties appear of similar magnitude across altitudes, and for D > 12 nm the uncertainties at lower altitudes are even smaller than those at higher altitudes.

*We rephrased the lines as follows:*

*At lower altitudes, particularly at high latitudes, the median values* *show less robust results* *due to limited data availability.*

- Lines 238-240: This result is expected, since the tropopause is generally higher at mid-latitudes than at high latitudes. You may consider making this explicit in the text to clarify why cirrus are found at higher absolute altitudes in mid-latitudes, but at similar relative altitudes with respect to the tropopause.

*We thank the reviewer for this suggestion and included its remark in those lines:*

*Statistically, cirrus clouds at mid-latitudes were observed at higher altitudes than those at high latitudes, as shown in Fig. 2 (d). However, the altitudes observed relative to the tropopause were similar for both latitude ranges.* *This result is expected, since the tropopause is generally higher at mid-latitudes than at high latitudes (Dessler, 2009). In fact, the altitudes observed relative to the tropopause were similar for both latitude regions.*

- Lines 239-240: The authors note sparse high-latitude data below 225 hPa. Please clarify that this is largely due to the lower tropopause at high latitudes, so these levels correspond to the lower stratosphere where cirrus are rare, rather than just a limitation of the dataset.

*We added an extra sentence at the end of the paragraph:*

*Given the lower tropopause heights at high latitudes, this range corresponds to the lower and drier stratosphere, where cirrus are rare.*

- Lines 243-245: The statement that high-latitude cirrus contain fewer but larger ice particles appears to be a deduction based on the similar IWC despite lower Nice. Please clarify that this is an inferred conclusion from Fig. 2, or provide explicit evidence from the particle size distribution to support it.

*We included an additional panel showing also a vertical profile of Dice, which is the evidence of this statement. We also included the citation of De La Torre Castro et al. (2023) and Jurkat-Witschas et al., (2025) where this finding is described and explained in more detail.*

- Lines 308-309: In the text, it is stated that "Climatological averages of these properties for June and July over the period 2014-2021 confirm this trend (see Fig. S2) and also serve as reference to contextualize the specific CIRRUS-HL episode." However, Fig. S2 actually shows a **model climatology**, not observational data. It would be helpful to clarify in the main text that the climatological reference is based on model output, averaged over the spatial domain of the flights, rather than on observations. Additionally, it would be valuable to include a comparison with observations from satellite or other observational climatologies for variables such as Nice, IWC, and RHice. This would help place the CIRRUS-HL observations into a broader context.

*We clarified that the climatology reference is based on model output. As a further reference, Fig. S4 also shows a comparison with the previous campaign ML-CIRRUS for both model and observations, allowing to place the CIRRUS-HL observations in a broader context and make the*

link to Righi et al., (2020), where also the model was compared with a larger observational climatology. We also extended Fig. S7 in the Supplement where we compare the simulation results with an observational climatology derived from various flight campaigns (Krämer et al., 2016, 2020) and also show IWC, Rice, and RHice together with Nice compared against the observations. We considered a comparison with MSG satellite observations, however, the nature of the flown patterns did not allow for sufficient data for comparison, since most of the measurements were not performed directly at the upper cloud level and without clouds below. Therefore, we did not pursue further this approach. A recent study by Li and Groß (2025) uses lidar measurements of CALIPSO focusing on the region of the CIRRUS-HL campaign to compare optical properties of high and mid-latitude cirrus. They also found differences in the extinction coefficients (larger) and optical depths (larger) for mid-latitude cirrus compared to high-latitude cirrus.

- Lines 311-313: The manuscript states that "the observed ice properties at mid-latitudes in the cold regime (< 220 K) are mostly well captured in the model," and that "at the lowest temperature of the measurements (≈ 212 K), the model predicts Nice and RHice comparable to the observations but overestimates IWC." Since IWC is overestimated across all temperatures and remarkably in mid-latitudes, this seems somewhat contradictory. It would be helpful if the authors clarify that, while Nice and RHice are reasonably reproduced, IWC is systematically overestimated, to avoid potential misinterpretation.

We adjusted the sentence as follows: *at the lowest temperature of the measurements (≈ 212 K), the model predicts Nice and RHice comparable to the observations but, as at warmer temperatures, it systematically overestimates IWC.*

- Lines 357: missing "at high latitudes" after "resulting cirrus" and before "(M-H)".

Noted and included, thank you!

- Lines 371-373: The final sentence beginning with "Ngo et al. (2025)" feels somewhat disconnected from the preceding discussion. I recommend adding a short transition or explanation to clarify how the findings of Ngo et al. relate to the results described here.

Thank you for the recommendation, we added the following sentence before that last statement to introduce it better: *Generally, the interplay of ambient conditions and the freezing efficiency of INPs is more important than total aerosol availability.*

- Lines 376-378: The authors state that "ice crystal number concentrations and the frequency of events are very likely overestimated, as suggested by recent findings by Testa et al. (2024)." However, Testa et al. (2024) primarily investigate the poor ice-nucleating efficiency of aviation soot particles and do not directly demonstrate an overall overestimation of ice crystal numbers or event frequencies. I recommend revising this sentence to more accurately reflect the scope of the Testa et al. study, or clarifying how their findings support this interpretation.

Thank you for this comment. We referred concrete to ice crystal number concentrations and frequency of events from aviation soot. We reformulate the sentence as follows to make it clearer:

*It is important to note that the ice crystal number concentrations and the frequency of freezing events from aviation soot ice-nucleation are very likely overestimated, as suggested by recent findings by Testa et al. (2024).*

- Lines 378-383: The paragraph discussing the potential impact of BCav as an INP is conceptually relevant but somewhat difficult to follow in its current form. The logical

connection between BCav efficiency, cirrus formation at high latitudes, and the vertical distribution of black carbon concentrations could be clarified.

*We reorganized the paragraph and added some details for clarification:*

*This difference arises because BCav and BC from ground sources differ not only in their ice-nucleating properties but also in their spatial distributions, leading to different interactions with cirrus clouds and, consequently, different climate effects. Soot from ground sources is likely to be aged (i.e., coated) by the time it reaches the upper troposphere, which limits its ice-nucleating ability. While black carbon concentrations from ground sources decrease with altitude and needs efficient vertical transport mechanisms, aviation emissions create a distinct concentration peak of BCav at cruise altitudes (Righi et al., 2021; Beer et al., 2022), precisely where in situ cirrus form and evolve. This local source is continuously renewed through flight activities e.g. in the North Atlantic Flight corridor. H-H cirrus are mainly of in situ origin (≈ 90%, De La Torre Castro et al., 2023), which form and remain at high altitudes where the relative abundance of BCav compared to other INPs is higher. However, any potential impact would be limited if BCav is assumed to have low ice-nucleating efficiency.*

It would also be helpful to briefly explain why BC and BCav behave differently despite their similar chemical composition. Clarifying this distinction would provide readers with a clearer understanding of the physical basis of the argument.

*We thank the reviewer for this recommendation, which has been addressed in the paragraph adjustments of the previous comment. As mentioned above, the ice-nucleating properties of aviation soot are very likely overestimated compared to recent measurements but this helps to facilitate the analysis of the results presented here.*

Lines 418-419: The manuscript states that "fit1 seems to match the observations more closely, it may lead to an overcorrection of Nice." At present, it is not clear **how this potential overcorrection is determined**. Is it inferred from deviations at other temperatures, from known model biases, or from the comparison with alternative fits (fit2) and climatological data? I recommend that the authors clarify the basis for this statement, for example by providing quantitative evidence or further explanation, so that readers can better understand why fit1 may overestimate ice crystal number concentrations.

*This statement is made based on the nature of the different geometrical approach of the sizing method of the ice crystals in the optical array probes. We reformulated lines 419-422 to improve clarity:*

*Fit1 may lead to an overcorrection of Nice because the minimum enclosing circle method systematically overestimates particle volume and ice mass when ice crystals have irregular or complex shapes. Fit2, however, relies on the area-equivalent diameter to calculate the mean volume diameter, providing a more realistic approximation of the actual ice crystal volume. This makes fit2 the more accurate choice, particularly in conditions where non-spherical ice shapes are prevalent (Wu and McFarquhar, 2016).*

Figures 1, 4, and S1: I recommend that the authors include the upper bound of the measurement range in Figures 1, 4, and S1. Indicating the complete range will help readers interpret the variability shown and better understand the limits of the data for each instrument and particle-size range.

*We feel that stating a specific number for the upper cutoff is misleading because it depends critically on particle density, air speed, sample flow settings and other parameters and is therefore*

not a fixed number but rather a range. For particle number concentrations this exact size is also not relevant because the large particle sizes contribute very little to the total number if Aitken- and accumulation mode are included in the measured range.

Lines 447-459 and 484-486: The conclusion correctly emphasizes that differences in ice crystal number concentrations are not explained by total aerosol concentrations, but rather by the influence of specific INP types and their interaction with ambient supersaturation. I suggest that the authors highlight the need for further studies focusing on the roles of different INPs across regions and cloud types, rather than only total INP concentrations, to better understand regional differences in cirrus microphysics.

We agree. We added the following sentence in lines 484-486:

*Accurate measurements of ice-nucleating particles at the low temperatures typical of the cirrus regime remain a critical need to advance our understanding of ice nucleation mechanisms and their role in the climate system. In particular, studies clarifying the roles of different INP types across regions and cloud types are needed to better explain regional differences in cirrus microphysics.*

- Lines 450-452: The conclusion refers to "differences in total INP concentrations," but it should be noted that the study only reports total aerosol concentrations (and particles >500 nm as a proxy for INPs). The actual concentrations of all INP types are not directly measured. I suggest clarifying this point, as the statement about total INP differences may overstate the available observational evidence.

We clarified the point as suggested by the reviewer and modified the sentence in the following way:

*We attribute the difference in cirrus microphysical properties primarily to the varying influence of specific INP types with distinct freezing efficiencies and its interaction with the ambient supersaturation within the mid- and high-latitude regions, rather than to differences in total aerosol concentrations (and particles > 500 nm as a proxy for INPs).*

- Lines 457-458: consider adding references that support the following satements: "denser aviation activity at mid-latitudes" and "which increases ice number concentration and reduces the effective particle diameter".

Thank you. We added Teoh et al. (2024), Gierens (2012), Voigt et al. (2017), Schumann et al. (2017), and Bier and Burkhardt, (2022) as references.

**Referee #2**

Overall, the manuscript is well written and nicely structured. The descriptions of the results are easy to follow, and the summary section nicely highlights the key findings. The combined usage of observations and simulations enhanced the interpretation of the observed results, as well as providing a pathway to improve the model. The reviewer has some minor comments below and recommends that these questions be addressed before the manuscript can be considered for publication.

1. Size restrictions to ice particles when comparing observations and simulations

- In Section 2.1, the authors mentioned that 3 cloud probes were used to measure ice particles, including CDP (2-50 micron), CIPgs (15-960 micron), and PIP (100-6400 micron). Then the authors mentioned that "to obtain a complete particle size distribution, data from the three instruments were combined". The reviewer wonders how exactly these 3 probes were combined, such as X, Y, Z range provided by CDP, CIPgs, and PIP, respectively, since they have overlapping ranges.

We included the ranges for calculating the combined particle size distribution and referred to publications for more details:

*To obtain a complete particle size distribution, data from the three instruments were combined (CDP: 2 – 37.5 μm; CIPgs: 37.5–247.5 μm; average CIPgs-PIP: 247.5–637.5 μm; PIP: 637.5–6400 μm; further details can be found in De La Torre Castro et al. (2023) and Sect. 4.3 of De La Torre Castro (2024)).*

- Later in the result section, line 280, section 4.2, an important size restriction was mentioned, that is, "only ice particles with diameters smaller than 200 micron" are used to calculate cirrus properties. The authors mentioned that > 200 micron particles will be categorized as snow crystals in the model and assumed to be removed within one model time step. This size cutoff at 200 micron will remove a significant amount of ice particles that would be the main contributor to ice water content (IWC). Evaluating how the model represents these >200 micron particles would be quite important for the overall model performance evaluation. Even though the model removes these particles in one time step, the authors can still show from a statistical point of view, before removal, how the model snow category's IWC, Ni, and Di compare with those derived for > 200 micron particles in observations. In fact, the reviewer suspects that the model biases seen in these larger size range would suggest that the one-step removal is too fast, and the total IWC of cirrus may be underestimated by the model. This finding could be as significant as the model biases of Nice overestimation as discussed in the later section.

In the model, ice crystals larger than 200 um, typically formed by aggregation, are transferred to snow crystals which are assumed to be removed within one model time step by precipitation, melting, or sublimation (Levkov et al., 1992). This threshold is introduced to avoid model instabilities which may arise due to a too-fast sedimentation of large ice crystals (Kuebbeler et al., 2014). This threshold can thus not simply be increased as it would result in an unstable model simulation. An evaluation of the fraction of ice crystals that are removed in this way would be complex, as all different microphysical process affecting these snow crystals would need to be tracked to achieve comparability with the observations and is therefore beyond the scope of this study. However, we acknowledge the uncertainty introduced by this threshold and included this in the text: *This threshold, while necessary to avoid model instabilities which may arise due to a too-fast sedimentation of large ice crystals (Kuebbeler et al., 2014), introduces an additional uncertainty which should be explored in future studies.*

- If the combined observations provided 15 micron – 6400 micron of ice particle size range, the authors may compare 15 – 200 micron with the modeled ice category, and 200 – 6400 micron with the modeled snow category. Please be cautious of the larger size cutoff (i.e., 6400 micron), that is, the model size range should be trimmed at both ends to match the observations more closely. That is, instead of using model at 200 – infinity, please trim it to 200 – 6400 micron. Overall, the reviewer recommends that this size restriction method be added to Section 3.2 instead of only mentioned briefly in the result section.

For the comparisons with observed ICNC, we indeed filtered the simulation data according to the size range of the measurements (i.e. 3 um < D < 6400 um). However, the upper limit is likely not affecting the results much as these very large ice crystals are very rare.

- In addition to IWC and Ni, can the author also quantify Dice (such as number-weighted mean or mass-weighted mean diameter) in Figures 2 and 5? Previously, observation-model comparisons such as Patnaude et al. 2021, Maciel et al., 2023, and Ngo et al., 2025 included the statistical distributions of IWC, Ni, and Di in midlatitudes and high latitudes as well. A quick comparison of CIRRUS-HL results with these previous observations obtained from other geographical regions would be helpful to understand the regional variabilities. In those previous studies, CESM2/CAM5 climate model was found to have too many but too small ice crystals (i.e., modeled Nice is too high and Dice is too low). It would be interesting to see if EMAC/ECHAM5 has a similar issue.

Thank you for this suggestion. We have included analyses regarding ice crystal sizes in addition to Nice and IWC. On the one hand for the CIRRUS-HL campaign, extending the overview in Fig. 2 of the 1-Hz time resolution of the ice microphysical properties and also comparing the simulated model data against the observations from the CIRRUS-HL campaign included in the new Fig. S3 (as in Fig. 5), and finally, against the observational climatology compiled by Krämer et al. (2016, 2020) (Fig. S7) on the other hand. We note that the ice crystal mean volume radius in the model is not an independent quantity as it is calculated from the simulated IWC values, assuming spherical particles. The results show an underestimation of the simulated crystal sizes compared to the observations and an overestimation in ice crystal numbers. These findings agree with a similar issue reported by Patnaude et al. (2021) and Maciel et al. (2023) for the NCAR CAM6 model, as the reviewer suggested.

2. Size range of aerosol measurements: In the comparison between observations and simulations, the authors used three ranges, i.e., D > 12 nm, > 14 nm, and > 250 nm (such as Figures 1 and 3). The usage of D > 250 nm seems to be due to the aerosol probe size range, although typically the D > 500 nm has been used as a proxy for INPs, like the references that authors cited. The reviewer wonders if there is a specific reason that the authors chose to focus on the > 250 nm instead of > 500 nm (even though > 500 nm was briefly discussed in Fig. S1). For instance, is there a concern of the data quality of > 500 nm, which is why the main analysis focuses on > 250 nm?

We thank the reviewer for this comment. We originally used D > 250 nm to adjust to the complete OPC size range. However, as pointed by the reviewer, we acknowledged the higher importance of the D > 500 nm size range used as proxy for INPs and decided to include it in the Supplement. However, there is no particular reason for not using D > 500 nm directly, and agreed to change the size range of Figs. 1 and 3 from D > 250 nm to D > 500 nm, and adjusted the text accordingly where necessary. We kept the comparison with D > 250 nm in the Supplement as before.

3. The reviewer assumes that the aerosol measurements used in this study included both in-cloud and clear-sky conditions. Thus, the reviewer recommends that a sensitivity test being conducted to verify if using only the aerosol measured at clear-sky conditions (excluding in-cloud aerosols) would produce similar main conclusions. Even though at cirrus temperatures, the interference of cloud hydrometeors on aerosol measurements is likely smaller than compared with that at higher temperatures, a data quality check on the aerosol measurements would still help to verify whether or not the aerosol number concentrations are affected by shattering of ice particles.

The reviewer's assumption is correct, the aerosol measurements in this study included both in-cloud and clear-sky conditions since our sensitivity study showed very little difference as shown in the following plot:

[Figure]

4. Thermodynamic/dynamic influences: The authors examined RHice in addition to cloud properties, and the reviewer recommends that the vertical velocity distributions (such as sigma_w, like the equation used to derive sigma_w for every 50 seconds or 500 seconds of observations in Patnaude et al., 2021; Ngo et al., 2025) also be examined besides RHice. In addition, it is interesting to see that the authors applied the back trajectory tool to separate the two origins of cirrus – liquid origin and in-situ, but these two types of cirrus were not investigated separately, such as in Figure 2 (just observations) and Figure 5 (comparison with simulations). It would be helpful to see if the two origins of cirrus also show significantly different microphysical properties, and if the model tends to show better or worse results for one type of cirrus. For example, if the model convective cirrus parameterization has been improved in Section 4.4, one would expect to see better result for the liquid-origin cirrus comparison between observations and simulations.

We acknowledge that increasing the granularity of the comparison and assessing separately liquid and in situ origin cirrus would be interesting. However, the approach in this study does not allow a robust comparison given the 15-min averages performed in the observations. Within the 15-min, and thus, within an EMAC grid box, we expect both liquid origin and in situ cirrus and therefore avoid to associate liquid or in situ cirrus averaged data points with the corresponding model grid box averages. In addition, this would further increase the scope of our current study and would be more suitable as a follow-up study or publication.

We included in Fig. S3 also a comparison of the updraft velocities between the model and the observations. Please, refer also to Fig. S3 of De La Torre Castro et al. (2023), where an overview of the vertical velocities observed during the CIRRUS-HL campaign was also shown but for 1-s time resolved observations.

5. The reviewer wonders if the model output variables used for comparisons are grid mean or in-cloud variables? Also, is the spatial averaging applied to both in-cloud and clear-sky segments of observations? It would be helpful to provide the model variable names and their meaning in section 3.1 or 3.2. In some previous studies of model evaluation such as Patnaude et al. (2021) and Maciel et al. (2023), the model grid-mean variables were used to compare with spatially averaged observations that included both in-cloud and clear-sky conditions. If one uses model variables that represent the in-cloud IWC and Ni, then those should be compared with in-cloud averaged IWC and Ni derived from observations.

Model output for the cloud properties are in-cloud values and in order to make it comparable, the averaging in the observations is also applied to in-cloud values. In addition to the description of the variables that have been used for both model and observations in the present manuscript, we also note that all the model variables included in the simulations are described in the data repository on Zenodo, which will be available after publication: https://doi.org/10.5281/zenodo.15534818

6. Spatially averaging the observations to 15 minutes seems a reasonable choice of scale. It is worth noting that in addition to the temporal frequency of the output, using similar spatial scales (i.e., model grid spacing scale versus the scale of the spatially averaged observations) is also important. Assuming that HALO flies at 250 m/s at the cirrus temperatures, 15 minutes * 250 m/s is about 225 km, which is comparable to the 2.8 deg * 2.8 deg lat * lon resolution of the model configuration as well. This similarity in spatial scale is worth noting too.

Thank you for noting this, we included a sentence in the first paragraph of Section 3.2:

*To match the model time step, we also average the observed aerosol and ice cloud properties over 15 minutes (an overview of the data before averaging is provided in Sect. 4.1). This averaging interval also provides a spatial scale comparable to the model grid resolution of 2.8 ∘ × 2.8 ∘, as an average cruise speed of 250 km results in a flight distance of 225 km in 15 minutes.*

7. In Figure 5, should the black dots in panels c and f, actually be orange color? Please clarify. It is a little confusing why the simulation data (orange) is broken up. Is it because of fewer vertical bins?

The dots mentioned by the reviewer are actually orange/red, at least as seen in the digital version. The simulation data is broken up due to the fewer data points at high latitudes, especially in the model. We mentioned it in the figure caption: *Discontinuous lines appear in the high-latitude panels due to insufficient statistics in those temperature bins.*

8. A minor comment on Figure 7's line format: The dotted and dashed lines have small gaps, which are readable on computer screen, but when printed out, the lines look identical within each panel. The lines also used colors that are too similar within each panel. The reviewer recommends using slightly different colors for various lines to enhance the readability of this figure.

Thank you for noting this, we adjusted the plot accordingly.

9. Line 38, "INPs, (Kanji et al.," has an extra comma after INPs.

Thank you! We corrected it.

**References**

Beer, C. G., Hendricks, J., Righi, M., Heinold, B., Tegen, I., Groß, S., Sauer, D., Walser, A., and Weinzierl, B.: Modelling mineral dust emissions and atmospheric dispersion with MADE3 in EMAC v2.54, Geosci. Model Dev., 13, 4287–4303, https://doi.org/10.5194/gmd-13-4287-2020, 2020.

Beer, C. G., Hendricks, J., and Righi, M.: Impacts of ice-nucleating particles on cirrus clouds and radiation derived from global model simulations with MADE3 in EMAC, Atmospheric Chemistry and Physics, 24, 3217–3240, https://doi.org/10.5194/acp-24-3217-2024, 2024

De La Torre Castro, E.: Microphysical properties and interplay of natural cirrus, contrail cirrus and aerosol at different latitudes, Phdthesis, Delft University of Technology, http://resolver.tudelft.nl/uuid:1daa70eb-e8ac-4ee7-ad53-3bd87c2de258, 2024.

De La Torre Castro, E., Jurkat-Witschas, T., Afchine, A., Grewe, V., Hahn, V., Kirschler, S., Krämer, M., Lucke, J., Spelten, N., Wernli, H., Zöger, M., and Voigt, C.: Differences in microphysical properties of cirrus at high and mid-latitudes, Atmospheric Chemistry and Physics, 23, 13 167–13 189, https://doi.org/10.5194/acp-23-13167-2023, 2023.

Dessler, A. E.: Clouds and water vapor in the Northern Hemisphere summertime stratosphere, Journal of Geophysical Research: Atmospheres, 114, https://doi.org/10.1029/2009JD012075, 2009.

Gierens, K.: Selected topics on the interaction between cirrus clouds and embedded contrails, Atmospheric Chemistry and Physics, 12, 11 943–11 949, https://doi.org/10.5194/acp-12-11943-2012, publisher: Copernicus GmbH, 2012.

Levkov, L., Boin, M., and Rockel, B.: Impact of primary ice nucleation parameterizations on the formation and maintenance of cirrus, Atmospheric Research, 38, 147–159, https://doi.org/10.1016/0169-8095(94)00091-Q, 1995.

Li, Q. and Groß, S.: Lidar observations of cirrus cloud properties with CALIPSO from midlatitudes towards high-latitudes, Atmos. Chem. Phys., 25, 16657–16677, https://doi.org/10.5194/acp-25-16657-2025, 2025.

Righi, M., Hendricks, J., Lohmann, U., Beer, C. G., Hahn, V., Heinold, B., Heller, R., Krämer, M., Ponater, M., Rolf, C., Tegen, I., and Voigt, C.: Coupling aerosols to (cirrus) clouds in the global EMAC-MADE3 aerosol-climate model, Geosci. Model Dev., 13, 1635–1661, https://doi.org/10.5194/gmd-13-1635-2020, 2020.

Righi, M., Hendricks, J., and Beer, C. G.: Exploring the uncertainties in the aviation soot–cirrus effect, Atmos. Chem. Phys., 21, 17 267–17 289, https://doi.org/10.5194/acp-21-17267-2021, 2021.

Schumann, U., Baumann, R., Baumgardner, D., Bedka, S. T., Duda, D. P., Freudenthaler, V., Gayet, J.-F., Heymsfield, A. J., Minnis, P., Quante, M., Raschke, E., Schlager, H., Vázquez-Navarro, M., Voigt, C., and Wang, Z.: Properties of individual contrails: a compilation of observations and some comparisons, Atmospheric Chemistry and Physics, 17, 403–438, https://doi.org/10.5194/acp-17-403-2017, 2017.

Teoh, R., Engberg, Z., Schumann, U., Voigt, C., Shapiro, M., Rohs, S., and Stettler, M. E. J.: Global aviation contrail climate effects from 2019 to 2021, Atmospheric Chemistry and Physics, 24, 6071–6093, https://doi.org/10.5194/acp-24-6071-2024, publisher: Copernicus GmbH, 2024.

Voigt, C., Schumann, U., Minikin, A., Abdelmonem, A., Afchine, A., Borrmann, S., Boettcher, M., Buchholz, B., Bugliaro, L., Costa, A., Curtius, J., Dollner, M., Dörnbrack, A., Dreiling, V., Ebert, V., Ehrlich, A., Fix, A., Forster, L., Frank, F., and Zoeger, M.: ML-CIRRUS: The Airborne Experiment on Natural Cirrus and Contrail Cirrus with the High-Altitude Long-Range Research Aircraft HALO, Bulletin of the American Meteorological Society, 98, 271–288, https://doi.org/10.1175/BAMS-D-15-00213.1, 2017.